# The role of refractive indices in measuring mineral dust with high-spectral resolution infrared satellite sounders: Application to the Gobi Desert

Perla Alalam[1,*,**], Fabrice Ducos[1], Hervé Herbin[1]

[1]Laboratoire d'Optique Atmosphérique, LOA, UMR 8518, CNRS, Université de Lille, F-59000 Lille, France
* now at: Univ. Paris Est Créteil and Université de Paris, CNRS, Laboratoire Interuniversitaire des Systèmes Atmosphériques, Institut Pierre Simon Laplace, Créteil, France
** now at: Centre National d'Etudes Spatiales, CNES, 75001, Paris, France

*Correspondence to*: Hervé Herbin (herve.herbin@univ-lille.fr)

**Abstract.** Mineral dust significantly influences the Earth's climate system by affecting the radiative balance through the emission, absorption and scattering of solar and terrestrial radiation. Estimating the dust radiative effect remains challenging due to the lack of detailed information on the physical and chemical properties of dust. High-spectral-resolution instruments in the infrared (IR) spectrum, such as the Infrared Atmospheric Sounding Instrument (IASI), have demonstrated the ability to quantify these aerosol properties. A crucial parameter for characterizing mineral dust from space is the complex refractive index (CRI), as it links the dust's physical and chemical properties to their optical properties.

This paper examines the impact of six prior laboratory CRI datasets to improve the characterization of dust microphysical properties using IASI. The CRIs include older measurements obtained through the classical pellet method, commonly employed in mineral dust applications, as well as newer datasets that incorporate the latest advancements in laboratory measurement techniques for aerosol generation. These datasets are tested on IASI measurements during a dust storm event over the Gobi Desert in May 2017. We evaluate the sensitivity of IASI to different CRI datasets using the ARAHMIS radiative transfer algorithm and explore their impact on retrieving size distribution parameters by mapping their spatial distribution. The results indicate that the laboratory CRI datasets decrease the total error of the covariance matrix by 30% . In addition, we assess the capability of accurately reconstructing IASI detections and the extent to which we can retrieve the microphysical properties of dust particles. The choice of CRI significantly impacts the accuracy of dust detection and characterization from satellite observations. Notably, datasets that incorporate recent aerosol generation techniques with higher spectral resolution and samples from the case study region show improved compatibility with IASI observations. The outcomes of this research emphasize two key points: the crucial link between chemical composition of dust and its optical properties, and the importance of considering the specific composition of the CRI dataset for improved retrieval of the microphysical parameters. Furthermore, this study highlights the critical role of ongoing enhancements in CRI measurement approaches, as well as the potential of high-spectral-resolution infrared sounders for aerosol atmospheric investigation and understanding their radiative impacts.

## 1 Introduction

Mineral dust, carried by strong winds from arid regions into the atmosphere, interacts with solar and terrestrial radiation, significantly impacting Earth's radiative budget (Kok et al., 2017). To date, estimating the aerosol radiative effect presents considerable uncertainties, as highlighted in reports by the Intergovernmental Panel on Climate Change (IPCC, 2023). These uncertainties primarily arise from the lack of detailed knowledge about the physical and chemical properties of mineral dust, which exhibit significant spatial and temporal variations (Masmoudi et al., 2003; Journet et al., 2014). The size distribution of dust

particles ranges from hundreds of nanometers to tens of micrometers, with the mineral composition varying according to the dust's source region (Ryder et al., 2018). This diversity allows remote sensing instruments to detect mineral dust by capturing spectral signatures in the ultraviolet (UV), visible (Vis), and infrared (IR) spectral ranges. Notably, satellite sensors offer a comprehensive

perspective on the mineral dust cycle, encompassing its emission, transport, and deposition, and spanning from regional to global scales . Consequently, radiative transfer models have been developed to simulate and reconstruct satellite observations. These models incorporate critical knowledge on the physical, chemical, and optical properties of dust particles.

In the last 20 years, IR instruments have made significant contributions to the detection of mineral dust due to their high sensitivity to the particles' composition. They offer the advantage of distinguishing the vibrational modes of various minerals and excel in

nighttime detections (Sokolik et al., 1998; Ryder et al., 2019). For example, high-spectral-resolution satellite sounders, such as the Infrared Atmospheric Sounding Instrument (IASI) and Atmospheric Infrared Sounder (AIRS), exploit the spectral variations of dust in the thermal IR range (750 - 1250 cm$^{-1}$) to quantify the physical and chemical properties of dust using radiative transfer algorithms (Pierangelo et al., 2005; Clarisse et al., 2010; Klüser et al., 2015; Capelle et al., 2018). Nevertheless, significant uncertainties persist in IR radiative transfer retrievals from satellite observations, primarily due to our limited knowledge of the

complex refractive index (CRI) of mineral dust. The CRI links the particles' optical properties, characterized by absorption and scattering processes, to their chemical composition or mineralogy, which vary from one source to another.

One of the earliest datasets for CRI applied to mineral dust originated from Peterson and Weinman, 1969 and focused on pure crystalline quartz, a major dust component. They exploited reflectance spectra and dispersion theory of solid crystal that were originally obtained by Spitzer and Kleinman, 1961. In the 1970s, the first natural dust samples, obtained from filtered rainout

precipitation and Saharan dust, were processed into glassy disc KBr pellets to measure reflectivity, and subsequently calculate the CRI in the IR spectrum (Volz 1972, 1973).

In the following years, recognizing that mineral dust consists of an aggregate of different minerals such as silicates, clays, carbonates, and iron oxides, various pure mineral CRIs were examined across a spectrum from infrared to ultraviolet using the pellets technique (Querry et al., 1978; Egan and Hilgeman, 1979; Glotch et al., 2007). Balkanski et al., 2007 further refined aerosol

radiative forcing assessments in the visible spectrum, employing pure mineral mixtures alongside literature CRIs, yielding more accurate radiative estimates than those by Volz 1972, 1973 . Furthermore, Capelle et al., 2014 studied the sensitivity of the IASI brightness temperature to the change in CRI. By comparing Volz 1972, 1973 to the "revisited" mineral dust CRI by Balkanski et al., 2007, the results showed high impact on the radiative transfer model.

From the 2000s, aerosol optical properties research increasingly incorporated advanced experimental techniques. These involved

aerosol generation methods that are more representative of natural airborne dust, in contrast to the pellet/film approach that alters particle size, shape, and vibrational modes. One of the initial systems to combine mineral dust generation with extensive measurement analysis, was established at the University of Iowa, utilizing the Multi-Analysis Aerosol flow Reactor System (MAARS) (Gibson et al., 2006). This setup was specifically used to generate clay mineral aerosols for extinction measurements and particle size distribution analysis, with a focus on examining the sphericity of the particles (Hudson et al., 2007). The

Rutherford Appleton Laboratory's Molecular Spectroscopy Facility (RAL-MSF) was also adapted for aerosol generation and high-resolution extinction spectrum measurements, alongside size distribution analysis to study the sphericity and crystallinity of quartz particles and derive the CRI of volcanic ash (Reed et al., 2017, 2018). Moreover, at the University Paris-Est Créteil, the CESAM (Chambre Expérimentale de Simulation Atmosphérique Multiphasique) was employed to generate mineral dust aerosols from various desert soils worldwide. These experiments, conducted under dry conditions, involved measuring extinction and size

distribution to obtain CRI (Di Biagio et al., 2017). Finally, at the University of Lille, the PhysicoChimie des Processus de Combustion et de l'Atmosphère (PC2A) platform has been employed for generating samples from the Gobi Desert, pure minerals

like quartz, illite, kaolinite and calcite and volcanic ash under dry atmospheric conditions, aiming to measure the extinction coefficient and size distribution and derive the CRI data spanning a wide spectral range from far infrared to UV (Deschutter, 2022; Deguine et al., 2023; Chehab et al., 2024).

The objective of this paper is to investigate the influence of the most recent CRI laboratory measurements with advanced generation and measurements techniques in comparison to previous datasets based on classical methods on IASI mineral dust retrievals. Firstly, we assess the sensitivity of IASI observations to a range of CRI datasets from literature. This includes a detailed analysis of the information content in these datasets, aiming to understand how variations in CRI measurements influence the IASI's ability to accurately detect and characterize the microphysical dust properties. This step is crucial for identifying the strengths and

limitations of current datasets while emphasizing the need for enhancements in CRI measurement techniques. Secondly, we apply the inversion process to a dust event that occurred on 4 May 2017 over the Gobi Desert in East Asia. By focusing on this specific event, we examine the capability of accurately reproducing detections and the extent to which we can retrieve the microphysical properties of dust particles. This analysis also evaluates how the incorporation of the most recent CRI measurements and the mixture methodology from the mineralogical study on East Asia by Alalam et al., 2022 can enhance the accuracy of mineral dust

microphysical properties retrievals.

**2 Case Study**

East Asia is the second largest source of mineral dust after the Sahara, producing up to 800 Tg per year (An et al., 2018). This significant output is primarily associated to cold fronts from Siberia and Mongolia. The significant temperature differences between cold Siberian air masses and warmer air to the south create strong pressure gradients, leading to high wind speeds that lift and

transport dust particles particularly during the winter and spring months (Wang et al., 2004). An intense dust storm occurred from 3 to 8 May 2017, affecting visibility across North China. From the southwest of Inner Mongolia, the dust plume travelled passing through North China, the Korean peninsula, and Japan before dissipating in Russia. Three cold fronts generated dust loads during this event. The first two fronts lifted dust from the Gobi Desert on 3 May. A third front emerged on 4 May, resulting in distinct dust plumes that merged between May 4 and May 6, originating from both the Horqin Sandy Land and the Gobi Desert (Minamoto

et al., 2018; Alalam et al., 2022). Due to cloud coverage during the dust event, we only focus on the 4 May, when the dust plume was at its maximum dispersion and visibility to the IASI observations.

To select IASI dust spectra, we employ the 'V-shape' dust criterion. As the dust concentration increases, the 'V-shape' slopes become more pronounced, making measurements in the atmospheric window (between 800 and 1200 cm$^{-1}$) more sensitive to concentration fluctuations (Sokolik et al., 1998 and 2002). Consequently, the difference in brightness temperature is a reliable

indicator for the AOD. Based on the differences in brightness temperature of the two 'V-shape' slopes we calculate $\Delta T_{B1} = T_{B,809.25} - T_{B,988}$ and $\Delta T_{B2} = T_{B,1191.25} - T_{B,1112}$, and apply a minimum sensitivity condition on both slopes of $\Delta T_B > 0.9$ K. Specific IASI channels were selected by empirically testing hundreds of dust spectra to minimize the effects of gas absorption lines. **Figure 1** illustrates the IASI difference in brightness temperature $\Delta T_{B2}$ during the Gobi dust storm on the 4 May 2017 at daylight hours. Only $\Delta T_{B2}$ was used for visualization to ensure accurate correction of surface emissivity. This choice is based on

its spectral range, where surface emissivity significantly affects the spectrum, as detailed by Alalam et al., 2022.

# 3 The datasets and methods

## 3.1 IASI data selection

IASI is an infrared Fourier transform spectrometer that is part of the METOP satellite series, developed by the CNES in cooperation with EUMETSAT. Since 2006, two sun-synchronized METOP satellites (B and C) are still in orbit, each with an IASI instrument on board. METOP A was switched off in November 2021, and a new generation of IASI-NG will continue the mission for the next 20 years, featuring increased spectral resolution and radiometric performance (Crevoisier et al., 2014). The instrument scans in nadir view with a swath of 2200 km. The field of view corresponds to $2 \times 2$ circular pixels, each with a 12 km diameter footprint at nadir. The IASI covers a continuous infrared spectral range between 645 and 2760 $cm^{-1}$ (3.62 and 15.5 µm) providing 8461 channels with a spectral resolution of 0.5 $cm^{-1}$ and low radiometric noise (Blumstein et al., 2004).

The mineral dust spectral selection method, as detailed by Alalam et al., 2022 is briefly summarized here. The input data are taken from IASI-A level 1c and level 2 data from EUMETSAT data center (https://data.eumetsat.int/). A land surface emissivity (LSE) correction method is applied to the IASI radiances to remove the LSE effect caused by the high variability of the surface emissivity especially, above deserts. Following this, the spectra are processed using a principal component analysis (PCA) code developed at the Laboratoire d'Optique Atmosphérique (LOA) by Herbin, 2014 and improved from the method of Atkinson et al., 2010. The PCA filters spectra into opaque, dust, cloud, and clear sky pixel types, with a requirement of 90% of type homogeneity in the IASI pixel. Mineral dust spectra are selected by the condition on the difference in brightness temperature mentioned in **Section 2**.

## 3.2 The CRI data

In this study, we choose to compare relevant CRI datasets employed in radiative transfer algorithms in the mid-infrared (MIR) spectral range as summarized in **Table 1.**, while further references can be found and are listed by Clarisse et al., 2019. The selected CRI datasets are as follows:

1- The first dataset originates from Volz, 1972 (VZ72), where the dust was separated from mid-latitude natural precipitation, including a mixture of soil particles, airborne soot, and pollen. This rainout precipitation, carries dust particles that are moved over long distances and deposited through a process called 'wet deposition'. In our study, we consider this dataset as it includes dust particles that have been transported over thousands of kilometers. The sample was blended and pressed under vacuum conditions to form a glassy disc using the classical pellet method. The CRI was calculated based on the measured absorption coefficient of the bulk material. In the MIR, its average spectral resolution is 50 $cm^{-1}$. It has previously been used in altitude and concentration retrievals from IASI (Vandenbussche et al., 2013; Capelle et al., 2017).

2- Similar measurements were conducted on the second dataset, also from Volz, 1973 (VZ73), with the sample collected from Saharan sand in Barbados, having an average spectral resolution of 10 $cm^{-1}$. This dataset has been widely utilized in Saharan dust retrievals (Clarisse et al., 2019; Desouza-Machado et al., 2010), as well as IASI dust level 2 products (ULB and LMD algorithms, https://cds.climate.copernicus.eu). Compared to other data from the Sahara, this dataset has shown to be the most representative for long-range transported Saharan dust (Clarisse & Astoreca, 2021).

3- The third dataset is derived from the Optical Properties of Aerosols and Clouds (OPAC) known as "mineral transported," calculated by Hess et al., 1998. It combines data from VZ73 and quartz with an average spectral resolution of 10 $cm^{-1}$. It has previously been used for quantifying Saharan and Asian dust (Cuesta et al., 2015; Klüser et al., 2015).

4- The fourth dataset, from Di Biagio et al., 2017 (DB17), represents a more recent measurement of refractive indices and is based on 19 global dust samples, with the soil collected from the Gobi Desert. Aerosols were generated and suspended in the CESAM chamber under dry conditions, and the CRI was retrieved using an optical inversion procedure with a

spectral resolution of 2 cm$^{-1}$. A recent study compared the Gobi and the Taklamakan Deserts' refractive indices with Volz in 1972, demonstrating a clear impact on the brightness temperature spectrum simulations in the MIR (Bi et al., 2020). These datasets were used in a climatological analysis of coarse-mode dust over global oceans (Zheng et al., 2023).

5- The fifth dataset, as measured by Deschutter, 2022 (DSC22) at the PC2A platform, also involves aerosols generated from a Gobi Desert dust sample but under dry atmospheric conditions with a spectral resolution of 1 cm$^{-1}$. The CRI was retrieved using an optimal inversion method, and the dataset was previously used by Alalam et al. in 2022 to determine the mineralogical fraction of East Asian dust from IASI spectra.

6- A sixth dataset consists of a mixture of pure minerals (quartz, illite, and calcite), as measured by Deschutter, 2022 with a spectral resolution of 1 cm$^{-1}$. We calculated an effective CRI using the Volume Mixing Approximation (VMA) one of the simplest approaches of the effective medium theory. The effective properties of an aggregate of minerals is considered as a weighted average of the properties of its pure minerals' constituents, with the weighting factors being their volume fractions (for a detailed explanation, refer to Sokolik and Toon, 1999). The volume fractions values are 15.3%, 80.0%, 4.7% for quartz, illite and calcite respectively. The percentages were determined based on linear combination calculations by Alalam et al., 2022, using the experimental extinction coefficient of the Gobi sample from which DSC22 CRI was derived.

**Figure 2** illustrates the complex refractive indices in the MIR range. Spectral signatures variations are apparent, primarily due to differences in the sampled dust regions, and therefore differences in the mineralogy. The most pronounced contrast in the CRI can be observed for VZ72, which originates from rainout precipitation rather than desert dust. DB17 and DSC22, both sampled from the Gobi Desert, exhibit similar spectral features but with distinct values, indicating that the setup and method used can also influence the derived CRI. However, the mineralogical variation of the soil sample collection region has a significant impact as well. On the other hand, OPAC is derived from a mixture of VLZ73 and quartz and has a greater impact on the real index than the imaginary index. Notably, the molecular signatures of quartz are more prominent in the vicinity of 800 and 1100 cm$^{-1}$.

### 3.3 The inversion process

The inversion process follows the formalism of the optimal estimation method (OEM) described by Rodgers, 2000. This approach allows us to assess the sensitivity of the measurement and its information content, and the separation of parameters derived directly from the measurement and those provided by the a priori state. Furthermore, it enables the calculation of errors arising from measurement noise and the smoothing effect imposed on the actual profile by the observation system, which includes both the instrument measurement and the non-retrieved parameters.

### 3.3.1 The forward model

To solve the forward transfer equation, we use an analytical relationship that links between the set of observations $\boldsymbol{y}$ (in this case, the IASI radiances), and the state vector $\boldsymbol{x}$, which its elements consist of the variables to be retrieved, and it is written as:

$$\boldsymbol{y} = \boldsymbol{F}(\boldsymbol{x}, \boldsymbol{b}) + \boldsymbol{\varepsilon} \tag{1}$$

where $\boldsymbol{F}$ represents the forward model $\boldsymbol{b}$ correspond to the fixed parameters affecting the measurement (i.e., the atmospheric conditions, gases concentrations, surface emissivity and temperature) and $\boldsymbol{\varepsilon}$ is the measurement error vector.

For this purpose, we employ the ARAHMIS code, a line-by-line radiative transfer code developed at LOA. This code has been previously used by El Kattar et al., 2020 to characterize greenhouse gases (CO$_2$ and CH$_4$) using the ground-based high-spectral-resolution infrared instrument CHRIS. Moreover, it is currently employed as a reference code for the preparation of space missions such as IASI-NG, HYSP, and MicroCarb.

In this work, the state vector $\boldsymbol{x}$ elements are:

(1) The geometric mean diameter $D_g$, which follows a lognormal distribution with a geometric standard deviation $\sigma_g$. To compare results with literature, we calculate an effective diameter ($D_e = D_g e^{2.5(ln\sigma_g)^2}$) corresponding a fixed geometric standard deviation $\sigma_g = 2.0$ µm as suggested by Clarisse et al., 2019. This values falls within the typical range measured for dust aerosols between 1.75 and 2.25 µm (Reid et al., 2003). Note that for other types of particles, this assumption may not be valid.

(2) The volume mixing ratio (VMR), which is defined as the ratio of the volume of a dust with respect to the total volume of the air sample. It expressed in parts per million (ppm) providing a standardized way to express the aerosols concentration in the atmosphere.

We chose only these two elements to establish a link between the microphysical properties of dust (size and concentration) and their mineralogical composition identified in Alalam et al., 2022, for the same case study. The forward model is computed using the ARAHMIS code, allowing for precise simulations of observed IASI radiances across the MIR, from which we select spectra ranging from 785 to 1235 cm⁻¹. The atmosphere is discretized into layers, each with a 1 km thickness. Gaseous spectroscopic parameters, including spectral line positions, intensities, and half-maximum widths, are computed based on the updated HITRAN 2020 database. The atmospheric conditions, such as pressure, temperature, and water vapor profiles, were derived from the UWYO database at the Dalanzadgad station (http://weather.uwyo.edu/upperair/sounding.html). The ozone profile was obtained from the WOUDC database at the Xhianghe station (https://woudc.org). The $CO_2$ and $CH_4$ profiles were taken from the CAMS Greenhouse Gases reanalysis (https://ads.atmosphere.copernicus.eu). Surface temperature data were acquired from the level 2 IASI product provided by EUMETSAT (https://data.eumetsat.int/). Nevertheless, it's essential to note that in this case study the IASI detections are primarily over land areas, and can be highly affected by the surface emissivity variability. To address this, we use the Zhou et al., 2014 datasets and apply the land surface emissivity (LSE) method demonstrated by Alalam et al., 2022 to reduce the variability effect before the selection process. A correction factor is calculated using the Reststrahlen feature criterion, which is found in both the emissivity spectra and IASI radiances of desert surfaces. This approach helps to correct the surface temperature and emissivity differently for each observation.

For aerosols, we consider the single scattering approximation, which assumes that the distance between aerosol particles is larger than the range of their size distribution. Consequently, the particles are sufficiently spaced such that the scattering of light by one particle occurs independently of others. This assumption is applicable in our case where we select only non-opaque IASI spectra using the PCA code. To calculate the aerosols extinction coefficient $k_{ext}^{aerosols}$, we use a Mie scattering code designed for spherical particles. This approach was previously used in the MIR as it is less sensitive to the shape of particles (Yang et al., 2007; Di Biagio et al., 2017). Since it is difficult to differentiate between fine and coarse modes in the MIR spectrum, a single median size distribution mode with a wider standard deviation can be used. The particle size distribution is assumed to be monomodal and lognormal, similar to the method described by Pierangelo et al., 2005. This distribution is characterized by the total number of particles $N_0$, a geometric mean diameter $D_g$ and a geometric standard deviation $\sigma_g$. The mean layer dust altitude is set at $Z = 2$ km with a thickness of $L = 1$ km. This choice is based on the lidar CALIOP/ CALIPSO track orbit above the dust plume on May 4, 2017, as illustrated by Alalam et al., 2022.

**Figure 3** illustrates a comparison of the extinction coefficient as derived from the six CRI datasets using Mie theory and assuming an effective diameter $D_g$=1.0 µm. The extinction was normalized to remove the dependency on the concentration. All six dataset exhibit a characteristic signature associated with silicates, the familiar bent observed in the neighbouring of 1050 cm⁻¹. Deguine et al., 2020 highlighted that as the silica fraction increases, the extinction peak values move to higher wavenumbers. DB17, DCS22, and VMA are smoother and have more pronounced peaks: similar double peak tectosilicates feature is observed near 778 and 795 cm⁻¹, carbonates peak at 879 cm⁻¹ and phyllosilicates peak at 916 cm⁻¹. While a contrast in peaks intensities reflects the difference

in mineralogy between DB17 and DSC22. Notably, the spectral resolution is low in other datasets: in OPAC no double peak features can be distinguished, while VZ72 does not show any discernible peaks between 750 and 980 cm$^{-1}$, and VZ73 only exhibits a signature near 916 cm$^{-1}$. This difference may be attributed to the improvement in the experimental measurement in the late years,

the spectral resolution and the chemical composition of the dust source.

### 3.3.2 Information content analysis

The information content (IC) analysis enables us to establish the sensitivity of the inversion for each parameter sought, and hence make an optimal selection of those parameters and the constrains applied depending on their sensitivity on the spectrum. This analysis allows to quantify the impact of each parameter on the retrieval accuracy and allows us to gain in computational time and

increase the quality of adjustment, by constraining parameters and hence avoid too great correlation between them.

Following Herbin et al., 2013, two matrices ($\mathbf{A}$ and $\mathbf{S_x}$) can fully characterize the information provided by IASI and they are needed to perform this analysis.

The averaging Kernel matrix $\mathbf{A}$, gives the sensitivity of the retrieved state to the true state, and is given by:

$$\mathbf{A} = \delta\hat{x}/\delta x = \mathbf{GK} \tag{2}$$


where $\mathbf{K}$ is the Jacobian matrix written by $\mathbf{K} = \delta F/\delta x$ , and $\mathbf{G}$ is the gain matrix which rows are the derivatives of the retrieved state with respect to the spectral points, and is written by:

$$\mathbf{G} = \delta\hat{x}/\delta y = (\mathbf{K^T S_\varepsilon^{-1} K} + \mathbf{S_a^{-1}})^{-1} \mathbf{K^T S_\varepsilon^{-1}} \tag{3}$$

where $\mathbf{S_a}$ is the uncertainty covariance matrix on the knowledge of the prior state and $\mathbf{S_\varepsilon}$ the error covariance matrix of the forward model and the measurement.

Rodgers, 2000 showed that the trace of $\mathbf{A}$ represents the total degree of freedom for signal (dofs), that gives the number of independent pieces of information provided by the observing system as regards the state vector.

The knowledge of the state vector posterior to the measurement is described by the total error covariance matrix $\mathbf{S_x}$, and can be written as:

$$\mathbf{S_x} = \mathbf{S}_{\text{smoothing}} + \mathbf{S}_{\text{meas.}} + \mathbf{S}_{\text{fwd.mod.}} \tag{4}$$

where $\mathbf{S}_{\text{smoothing}}$ is the smoothing error covariance matrix and describes the vertical sensitivity of the measurements to the retrieved profile, and it is given by:

$$\mathbf{S}_{\text{smoothing}} = (\mathbf{A} - \mathbf{I})\mathbf{S_a}(\mathbf{A} - \mathbf{I})^\mathbf{T} \tag{5}$$

$\mathbf{S}_{\text{meas.}}$ is the contribution of the measurement error covariance $\mathbf{S_m}$ associated with spectral noise, and it is written as:

$$\mathbf{S}_{\text{meas.}} = \mathbf{G S_m G^T} \tag{6}$$

$\mathbf{S}_{\text{fwd.mod.}}$ is the contribution of the forward model error covariance matrix $\mathbf{S_f}$ associated with uncertainties from non-retrieved model parameters described by the covariance matrix $\mathbf{S_b}$:

$$\mathbf{S}_{\text{fwd.mod.}} = \mathbf{G K_b S_b (G K_b)^T} = \mathbf{G S_f G^T} \tag{7}$$

where $\mathbf{K_b}$ is the forward model derivative as regards non-retrieved model $\mathbf{x_b}$ and $\mathbf{S_b}$ is the uncertainty covariance matrix attached to $\mathbf{x_b}$.

### 3.3.2.1 A priori error covariance matrix

The a priori error covariance matrix $\mathbf{S_a}$ is assumed diagonal with the $i^{th}$ diagonal element ($\mathbf{S}_{a,ii}$) defined as:

$$\mathbf{S}_{a,ii} = \sigma_{a,i}^2 \text{ with } \sigma_{a,i} = x_{a,i}.\frac{p_{error}}{100} \tag{8}$$

where $\sigma_{a,i}$ is the standard deviation in the Gaussian statistics formalism. The subscript $i$ represents the $i^{th}$ parameter of the state vector. The prior knowledge of aerosol parameters ($D_g$, VMR) is supposed to be known with an uncertainty of 100 % (Frankenberg et al., 2012).


### 3.3.2.2 Measurement error covariance matrix

The measurement error covariance matrix is influenced by the radiometric calibration and the radiometric noise, given by the signal-to-noise ratio (SNR). This error covariance matrix is also assumed to be diagonal, and the $i^{th}$ diagonal element can be computed as follows:

$$\mathbf{S}_{m,ii} = \sigma_{m,i}^2 \text{ with } \sigma_{m,i} = \frac{y_i}{SNR} \tag{9}$$

where $\sigma_{m,i}$ is the standard deviation of the $i^{th}$ measurement ($y_i$) of the measurement vector $y$, representing the noise equivalent spectral radiance. In the case of the IASI instrument the SNR in the mid-infrared is set to 500 as the noise is stable within the MIR range and is equal to $2 \times 10^{-4}$ W.m$^{-2}$ sr$^{-1}$ (cm$^{-1}$)$^{-1}$ (Clerbaux et al ., 2009).

### 3.3.2.3 Non-retrieved parameters characterization and accuracy

For the temperature profile and surface temperature, we assumed a realistic uncertainty of 1 K, compatible with the typical values used for the IASI instrument, on each layer of the temperature profile as well as on surface temperature (Pougatchev et al., 2009). The contribution to the $i^{th}$ diagonal element of the forward model error covariance matrix from the $j^{th}$ level temperature can be computed as:

$$\sigma_{f,T_j,i} = \frac{\delta F_i}{\delta T_j}\Delta T \tag{10}$$

where $j$ stands for the $j^{th}$ level and $i$ for the $i^{th}$ measurement.

The surface emissivity ($\varepsilon_s$) uncertainty is set to $p_{\varepsilon_s} = 2\%$, as estimated by Capelle et al., 2012. The surface emissivity's contribution to the $i^{th}$ diagonal element of the forward model error covariance matrix is:

$$\sigma_{f,\varepsilon,i} = \frac{\delta F_i}{\delta \varepsilon_s}\Delta\varepsilon_s, \text{ with } \Delta\varepsilon_s = \varepsilon_s.\frac{p_{\varepsilon_s}}{100} \tag{11}$$

Another parameter that was not retrieved is the molecular gas concentration, $C_{mol}$ (in ppm). $H_2O$ is presumed to have an a priori error on the concentration profile of $p_{H_2O}=10\%$. This error value is compatible with the a posteriori uncertainty from IASI Level 2 products given by Clerbaux et al., 2007. The $CO_2$ uncertainty is set to $p_{CO_2} =1\%$ from Engelen and Stephens, 2004 while the $O_3$ error is $p_{O_3} =5\%$ (Boynard et al., 2016). The $CH_4$ error is estimated to be 5%, which is compatible with the estimation of De Wachter et al., 2017. For the other interfering molecule concentrations ($N_2O$, cfc-11 and cfc-12), we consider a weak prior

knowledge, and their uncertainties are fixed to 100 %.

The prior contribution to the $i^{th}$ diagonal element of the forward model error covariance matrix can be computed as:

$$\sigma_{f,C_{mol},i} = \frac{\delta F_i}{\delta C_{mol}}\Delta C_{mol}, \text{ with } \Delta C_{mol,k} = C_{mol}.\frac{p_{C_{mol}}}{100} \tag{12}$$

The uncertainty percentages are summarized in **Table.2**.

Finally, the total forward model error covariance matrix $\mathbf{S_f}$, assumed diagonal in the present study is given by the sum of all error

contributions for each diagonal element, and the $i^{th}$ diagonal element $\mathbf{S}_{f,ii}$ is given by:

$$\mathbf{S}_{f,ii} = \sum_{j=1}^{n_{level}} \sigma_{f,T_j,i}^2 + \sigma_{f,\varepsilon,i}^2 + \sum_{k=1}^{n_{mol}} \sigma_{f,c_{mol,k},i}^2 \tag{13}$$

Here, we did not consider the spectroscopic line parameter, line-mixing, continua or calibration errors.

## 4 IASI IC analysis: CRI evaluation

In this section, we study the impact of the laboratory CRI measurements to extract information on microphysical aerosol parameters: the geometric diameter $D_g$ and VMR. Accordingly, an IC analysis was performed for the elements of the state vector separately, considering the IASI spectral range between 785 and 1235 cm$^{-1}$, where mineral dust is detectable. We compute the extinction coefficient ($k_{ext}$) at 1020 nm. The AOD value at 1020 nm is often used as a reference because this wavelength is included in AERONET network data and facilitates easier comparisons with climatological studies e.g., Dubovik et al., 2002 showing that the regression of the optical parameters with 1020 nm are more robust. To avoid saturation the $AOD$ values at 1020 nm must be less than 2.00 and to avoid loosing sensitivity it must be greater than 0.05. Subsequently, we derive the VMR at the $AOD$ values interval (0.25 - 1.50), and it is given by:

$$VMR = \frac{AOD}{C_{air}\, k_{ext}\, L} \tag{14}$$

where $C_{air}$ (in ppm) is the concentration of the air in the atmospheric profile and $L$ is the layer thickness fixed by 1 km (as mentioned in the **Section 3**.

**Figure 4** illustrates the dofs and total error from the state vector parameters $D_e$ and VMR separately as function of the $AOD$ and $D_e$ between 1.5 and 5.0 µm . For all CRIs, the dofs is typically greater than 0.50, indicating that the information on mineral dust comes mainly from the measurement **y**. An exception is VZ72 at $D_e = 1.5\ \mu m$, which exhibits the worst case with a $dofs_{VMR}$ of 0.31 for an $AOD = 0.25$. Conversely, the best case is observed for VMA, where the $dofs_{D_e}$ is 0.99 for an $AOD = 1.00$. As the $AOD$ increases, the $dofs_{D_e}$ and $dofs_{VMR}$ increase and tend to 1, which indicates that the observation system should adequately provide the necessary information to derive $D_e$ and *VMR*.

On the other hand, the results suggest that at given $D_e$, all CRI errors $S_{x,D_e}$ and $S_{x,VMR}$ tend to decrease by 50 % as the $AOD$ increases from 0.25 to 1.50, showing a negative correlation of the total errors with the $AOD$. For instance, VZ72 errors at $AOD = 0.25$ are $S_{x,D_e} = 20\%$ and $S_{x,VMR} = 80\%$ ,while $AOD = 1.50$ the errors decrease to $S_{x,D_e} = 10\%$ and $S_{x,VMR} = 40\%$ . The CRIs have a similar trend when increasing $D_e$, $S_{x,D_e}$ and $S_{x,VMR}$ drop by 35%. For example, at $AOD = 1.00$, DB17 errors are $S_{x,D_e} = 12\%$ and $S_{x,VMR} = 37\%$ at $D_e = 1.5\ \mu m$, while these values decrease to $S_{x,D_e} = 8\%$ and $S_{x,VMR} = 25\%$ at $D_e = 5.0\ \mu m$. However, OPAC deviates from this general trend and exhibits an opposite behaviour. While for VMA, no matter the size diameter, its errors remain the least affected. This implies that VMA's errors to changes in particle diameter stay minimal as the diameter grows. Moreover, VZ73 with a close behaviour to DSC22 with $S_{x,D_e}$ of 4% and 5% respectively at $AOD = 1.50$, shows a good compromise that gives low errors regardless of the diameter choice justifying its use so far in the most IR remote sensing applications. DB17 shows an improvement compared to VZ72, consistently with a gain of 30% in sensitivity for all diameters. In conclusion, the different CRIs behaviour's in response to changes in size diameter can vary significantly. Compared to all other datasets, VZ72 exhibits the highest errors, which are primarily attributed to its low spectral resolution, as previously illustrated in **Fig. 3**. The fewer spectral features and structures the CRI possesses, the lower its sensitivity, resulting in higher errors. While most CRIs show an increase in sensitivity with increasing $D_e$, exceptions like OPAC demonstrate that unique characteristics and behaviours exist within the set. In addition, East Asian dust are transported at low altitudes, hence when the $AOD$ value is very high, the thermal contrast between the dust layer and the surface temperatures limits the sensitivity of satellite observations. This

is the case of VMA at $D_e = 5.0$ µm, where the errors increase again at $AOD = 1.50$. These results highlight that the selection of an appropriate CRI is crucial in remote sensing retrievals.

**5 Solving the inverse problem**

In this study, instead of using the linear scale, the iterative process is refined by using the logarithmic scale since the order of magnitude can highly vary between the vector state parameters i.e., the diameter value (in µm) is approximately ten orders of magnitude greater than the VMR value (in ppm). This is also the case between the order of magnitude of the vector state **x** and the measurement **y**. The logarithmic scale compresses the range making it easier to analyze trends and patterns, especially in the case of large variation in order of magnitude. The iterative process is then refined by:

$$lnx_{i+1} = lnx_a + \left(\mathbf{K_i^T S_\varepsilon^{-1} K_i + S_a^{-1}}\right)^{-1} \mathbf{K_i^T S_\varepsilon^{-1}} \times [y - F(x_i) + \mathbf{K_i}(lnx_i - lnx_a)] \tag{15}$$

where in this case $\mathbf{K_i} = \delta F/\delta lnx_i$ and $\mathbf{S_a} = \boldsymbol{\sigma_a^2}$ with $\sigma_{a,i} = lnx_a \cdot \frac{perror}{100}$

In this section, the inversion method is used to quantify the microphysical dust parameters ($D_e$ and VMR) and measure the ability of the simulated spectra to reproduce the IASI measurement we calculate the Root Mean Square ($RMS = \sqrt{\Sigma(y_i - F(x_i))^2 / n}$, where $n$ is the number of the spectral channels of 1804). The higher the value of the RMS, the wider is the spread around the IASI

spectra and the less is the ability of the spectral fit to reproduce the IASI spectrum.

**5.1 Application to three spectral pixels**

To evaluate the impact of the six refractive indices datasets in reproducing the IASI dust measurements, we applied the inversion process to three observations from the Gobi dust event on 4 May 2017, and calculated the RMS as a measure of the capability to reproduce the IASI spectra. These observations are distinguished by their brightness temperature differences $\Delta\mathbf{T_{B2}}$: spectrum 1 at

(44.3°N, 119.1°E) with $\Delta\mathbf{T_{B2}}= 2.6$ K, spectrum 2 at (44.0°N, 119.4°E) with $\Delta\mathbf{T_{B2}}= 5.0$ K and spectrum 3 at (49.6 °N, 124.6°E) with $\Delta\mathbf{T_{B2}}= 6.9$ K. Each spectrum exhibits a different V-shaped slope, indicating varying AOD, thereby providing a rigorous test for the CRIs to accurately reproduce the spectral observation. **Figure 5** illustrates the spectral fits (in red) from ARAHMIS to IASI observations (in blue) in terms of brightness temperature (in K). A good fit would mean that the model is able to reproduce the observed data with high fidelity, which in turn implies that the model's assumptions and inputs (i.e., atmospheric composition,

temperature profiles) are accurate representations of the actual atmospheric conditions. For the spectrum 1, having a small difference in brightness temperature, the simulations yield high precision across all CRIs. This underscores the capability of the atmospheric model and the ARAHMIS code to accurately reproduce the measurements, thereby affirming the reliability of these tools in the case of minimal aerosol concentration. The ability to reproduce the spectral fits decrease for all CRIs while $\Delta\mathbf{T_{B2}}$ increase, nevertheless the extent of this reduction varies across different indices. As $\Delta\mathbf{T_{B2}}$ increases, the spectral fits from VZ72,

VZ73, and OPAC exhibit challenges in replicating observations, particularly between 780 and 980 cm$^{-1}$. Meanwhile, between 1100 and 1230 cm$^{-1}$, the VZ72 and DB17 fits encounter the most difficulties, whereas VMA fits primarily experience difficulties only between 950 and 980 cm$^{-1}$. Notably, DSC22 fits exhibit a more robust capability in reproducing observations. By comparing with **Fig. 3**, it becomes evident that CRIs with the highest extinction values within a specific spectral range tend to face greater challenges in reproducing observations accurately within that range. Therefore, the CRIs spectral resolution and features affect the

ability to reproduce spectral observations.

We also evaluate the impact of the CRIs on the dust microphysical properties retrieval. $D_e$ and VMR, along with the associated RMS in K and W·m$^{-2}$·sr$^{-1}$·(cm$^{-1}$)$^{-1}$, are presented in **Table 2**. Indeed, all datasets demonstrate a rise in RMS values progressing

with increasing $\Delta T_{B2}$, indicating a loss of accuracy between the spectral fits and IASI observations. The DSC22 and VMA datasets exhibit the highest accuracy with the lowest RMS values, followed by DB17 and VZ73, while OPAC and VZ72 show the highest RMS values. In addition, the VMA dataset shows consistency in the RMS values regardless the difference in brightness temperature. As we approach saturation, the spectra increasingly reflect thermal emission from the aerosol layer, leading to a loss of sensitivity due to reduced spectral variation. This makes it more challenging to accurately reproduce the spectra. The results demonstrates no matter the CRI dataset used, as the aerosol loading increases, our ability to accurately reproduce the IASI spectra decreases, as given by the increasing RMS values. Moreover, by increasing $\Delta T_{B2}$, most datasets exhibit a decreasing trend in the $D_e$ and an increasing trend for VMR values except for DB17. Notably, VMA, DSC22 and VZ73 display the smallest effective diameters in the neighbouring of 3.2 µm, while DB17 shows the highest effective diameters for an average of 5.7 µm.

## 5.2 Application to the full dust plume

A large dust plume was dispersed between southwest and northeast China within a 2000 km² area in 4 May 2017. We select 1447 IASI dust observations between 2 and 4 UTC (between 10 am and 12 pm in Beijing local time) using the PCA code and study the impact of the six CRIs to retrieve the microphysical parameters using ARAHMIS. **Figure 6** shows the maps of effective diameter, the VMR and the RMS of the fitted spectra. The mean value represents the central tendency of each parameter, while the standard deviation the degree of dispersion around the mean, providing insight into the variability within each CRI. From 1447 selected observations, the retrieval process had no rejected values. It is also important to notice that there is a very weak correlation between the RMS and the microphysical parameters $D_e$ and VMR, across different CRIs, which demonstrates that the inversion process does not bias the retrievals. Therefore, the output for the 1447 observations aligns with a gaussian distribution for $D_e$, VMR and RMS. An example of distribution of $D_e$, VMR and RMS corresponding for DSC22 dataset is shown in **Fig. 7**, other datasets histograms are found in the **supplementary material**. There is a significant variation of approximately 30% in RMS values, with the highest being for VZ72 and the lowest for DSC22. The optimal mean RMS is $2.1 \times 10^{-3}$ W·m$^{-2}$·sr$^{-1}$·(cm$^{-1}$)$^{-1}$, is associated with the DSC22 and VMA datasets for 1447 pixels. The radiometric noise for the IASI instrument is approximately $2\times10^{-4}$ W·m$^{-2}$·sr$^{-1}$·(cm$^{-1}$)$^{-1}$ in the mid-infrared region. Given the complexity of our radiative transfer model, which includes numerous non-retrieved parameters such as temperature, pressure, and gases concentrations, a mean RMS of $2\times10^{-3}$ W·m$^{-2}$·sr$^{-1}$·(cm$^{-1}$)$^{-1}$ is considered within an acceptable range. Similarity between DSC22 and VMA's RMS values shows consistency between the CRIs obtained by Deschutter, 2022 and the mineralogical composition reported by Alalam et al., 2022. However, the RMS values for DB17 are consistently lower. This can be caused by the influence of humidity since DB17 reported that small amounts of water vapor and $CO_2$ contaminated the dust spectra below 7 µm. In contrast, DSC22 and VMA were obtained from the PC2A platform, which uses a nitrogen purge to reduce gases content in the apparatus. This difference could account for the observed variations. DSC22 and DB17 datasets are derived from measurements generated from Gobi Desert samples, while the VMA set is a mixture of pure minerals calculated for the Gobi samples' mineralogy, as previously calculated by Alalam et al. in 2022. Note that, there are two regions in the neighbouring of (38.0°N; 104.0°E and 43.0°N; 112.0°E) where the RMS is slightly higher for all CRIs. This corresponds to finer size distributions for which the MIR is less sensitive, and this is confirmed by the smaller size diameter retrieval for all CRIs. When natural dust CRIs are excluded, the most favourable RMS is observed for the VMA dataset, suggesting that a CRI calculated from a pure mineral mixture can be used as a reliable proxy for the natural dust sample CRI. This is consistent with the hypothesis of heterogeneous mixture of pure minerals, as verified by Deschutter, 2022 for the DSC22 sample using the Scanning Electron Microscopy (SEM). Integrating new CRI measurements and the mixing methodology detailed in the mineralogical study by Alalam et al., 2022, has shown the potential to improve the measurements reproduction.

The distribution of retrievals across all plume pixels exhibits a uniform range of magnitude across all CRIs, despite variations in specific values. A factor of 1.6 is observed between the lowest and highest values for the mean effective diameter (lowest in VMA and highest in DB17) and inversely for the VMR. As both DB17 and DSC22 CRIs are obtained from dry Goby dust samples, the difference is mainly due to the variation in the imaginary part of the refractive index, which can arise from differences in the chemical composition of these samples. The standard deviation indicates significant variability in these parameters among different CRIs. DSC22 exhibits the largest standard deviation (i.e., variability) for the effective diameter, while VZ73 has the lowest. This contrast highlights the differences in particle size distribution retrieval between these two CRIs. On the other hand, VMA shows the largest distribution width for VMR, therefore a high degree of variability. Conversely, OPAC has the narrowest distribution, suggesting more uniform VMR values. This analysis reveals different degrees of variability in the effective diameter and VMR across various CRIs.

From a geographical aspect, for all CRIs the spatial distribution displays an increase in diameter from southwest to northeast China. Following the mechanism of the wind front, dust is blown with large diameters northeast particularly from the Horqin Sandy Land and floating dust with smaller diameters from the Gobi Desert. DSC22 has the widest range of effective diameter values. Notably, only VMA and DSC22 show high VMRs in the center of the plume that align with the wind front lines as illustrated by Minamoto et al., 2018.

To validate our results, we didn't find enough ground measurements within the dust plume event that are statistically representative. Nevertheless, two Sun–Sky Radiometer Observation Network (SONET) stations include measurements within the dust plume at the same date and time of the IASI observations: Beijing (40.005; 116.379) and Yanqihu (40.408; 116.674). The size distributions are retrieved using an inversion method as described by Li et al., 2018. In both stations, the mean effective coarse diameter retrieved was of 3.4 μm with a standard deviation of 1 μm. Our results show coherent effective size diameters that fall in the range of SONET retrievals, especially the DCS22 dataset, where the mean effective diameter for the overall observations was found to be 3.2 μm with a standard deviation of 1.4 μm. However, it remains challenging to compare two different observational and inversion methods in which IASI is a satellite-based instrument measuring the infrared spectrum, while SONET employs ground-based sun-sky radiometers in visible spectrum, therefore different sensitivity to dust detection.

Finally, through this case study, we have been able to quantify the impact of CRIs on the retrieval of the aerosol microphysical parameters (size and concentration) which play a significant role in estimating the dust radiative effect.

**Conclusion and perspectives**

This study provides valuable insights into the role of dust CRIs in the aerosol microphysical retrievals using infrared remote sensing, in particular IASI detections. It emphasizes the critical importance of selecting the appropriate CRI for accurately determining the microphysical properties of these particles. Initially, we use; the ARAHMIS radiative transfer algorithm to evaluate the IASI measurements sensitivity to various CRIs commonly used in previous studies: VZ72, VZ73, OPAC, DB17, DSC22 and VMA. The information content shows that the IASI measurements is able to accurately retrieve particle size and volume mixing ratio, particularly at higher AOD levels. Moreover, the selection of an appropriate CRI can decrease the total error by 30% which was shown to be the best for the VMA dataset. Hence, improvements in optical properties dust measurements are demonstrated significant potential in aerosol parameter retrieval. This progress is important for future atmospheric studies and applications that rely on CRI laboratory measurements that have higher reliability to satellite spectral observations.

The next step, we applied the retrieval process on a dust storm that occurred over China's Gobi Desert on 4 May 2017. By applying the inversion process on three contrasted spectra in terms of brightness temperature difference, VMA and DSC22 showed the most accurate spectral fits to the dust observations. For the overall dust plume results, the microphysical parameters have a uniform distribution across all different indices despite the spread of the values. The spatial distribution of aerosol retrieved parameters was plotted across the East Asian region. The distribution patterns vary for different CRIs, which reflect the influence of CRIs choice on the retrieval. A very weak correlation between RMS and the microphysical properties across different CRIs suggests that the ARAHMIS inversion process is reliable, as it does not produce errors across the range of retrieved parameters. The RMS is only significant in the case of small diameters, indicating lower sensitivity to this size range.

The RMS values of DB17, DSC22, and VMA are significantly better than those of VZ72, VZ73, and OPAC, highlighting improvements in retrieving CRIs from resuspended particles, particularly in reproducing IASI detections. The optimal mean RMS is linked to CRI datasets derived from the Gobi Desert. Also, this RMS value indicates that despite the challenges posed by non-retrieved parameters, our model is reasonably capable of reproducing the IASI observations. In future, more improvements can focus on refining the treatment of non-retrieved parameters. On the other hand, our comparison was limited by the availability of ground measurements, with data only from two SONET network sites. We acknowledge that compensating effects among different variables can also occur. This is particularly true for fine particles, to which the IR is less sensitive, leading to higher VM for these values. To quantify this effect accurately, it is essential to have a sufficient number of ground-based measurements that are statistically representative of the dust plume location to validate this compensating effect. We recognize the importance of comprehensive verification and will consider expanding our verification dataset in future studies as more data become available. This will help to further validate our findings and enhance the robustness of our results.

The results also suggest that the accuracy of reproducing IASI spectra is associated with the source of the CRI dust samples. In the absence of definitive dust CRIs, the optical properties of pure mineral aggregates can reliably reflect the regional mineral composition, as shown by low RMS values in such case. This opens the perspectives to better quantify not only the dust mineralogical composition as shown in Alalam et al., 2022, but also, to retrieve more precise microphysical properties through an enhanced understanding of the composition which is crucial for accurately assessing its radiative effects and its impact on climate and human health.

**Data Availability.** Data are available upon request by contacting the authors.

**Author contribution.** Conceptualization, P.A. and H.H.; algorithm development: F.D. and H.H.; validation, P.A., F.D. and H.H; formal analysis, P.A., H.H. and F.D.; investigation, P.A. and H.H.; writing—original draft preparation, P.A. and H.H.; writing—review and editing, P.A. and H.H.; supervision, H.H.; project administration, H.H.; funding acquisition, H.H. All authors have read and agreed to the published version of the manuscript.

**Funding.** This work is a contribution to the LabEx CaPPA project funded by the French National Research Agency under contract 'ANR-11-LABX-0005-01' and to the CPER research project CLIMIBIO funded by the French Ministère de l'Enseignement Supérieur et de la Recherche. The authors thank the Regional Council 'Hauts-de-France' and the European Regional Development Fund for their financial support for these projects.

**Competing interests.** The authors declare that they have no conflict of interest.

**Acknowledgements.** Special thanks to Lise Deschutter and Denis Petitprez for providing us with experimental Gobi dust complex refractive indices. We also thank Kaitao Li and Philippe Goloub for providing us with SONET data measurements. Finally, we would like to thank François Thieuleux for his valuable guidance in efficiently operating the ARAHMIS algorithm on the LOA cluster.

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

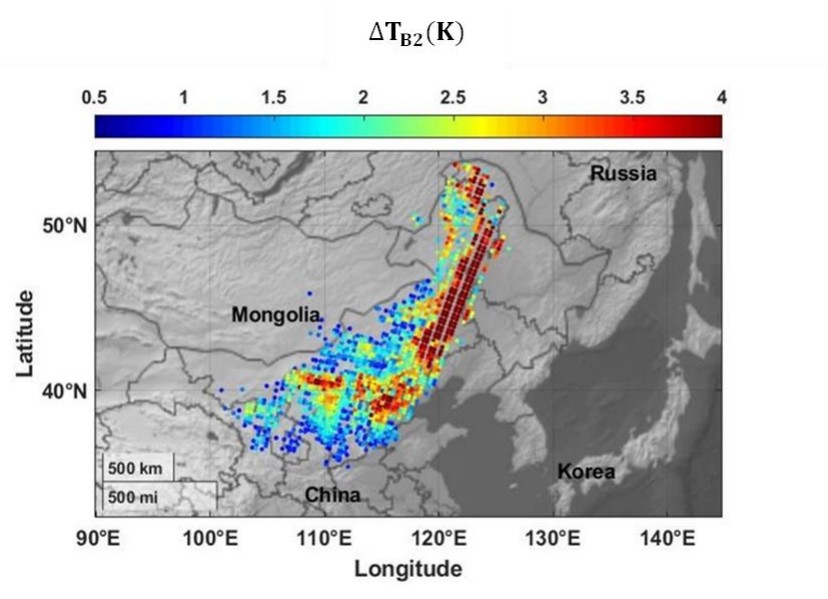

**Figure 1.** Difference in brightness temperature from IASI dust selection during the dust storm in 4 May 2017 at daylight.

**Table 1.** Summary of CRIs datasets used in this study.

| Reference | Abbreviation | Sample | Method | Total spectral range (cm⁻¹) | Spectral resolution in MIR (cm⁻¹) |
|---|---|---|---|---|---|

| Volz, 1972 | VZ72 | Mid-latitude rainout dust | Pellet | 250–50000 | ~50 |
|---|---|---|---|---|---|
| Volz, 1973 | VZ73 | Saharan sand, Barbados, West Indies | Pellet | 250–4000 | ~10 |
| Hess et al., 1998 | OPAC | Mixture of Volz,1973 and quartz | Pellet | 250–4000 | ~10 |
| Di Biagio et al., 2017 | DB17 | Gobi desert, China | Suspended aerosols | 666–3333 | 2 |
| Deschutter, 2022 | DSC22 | Gobi desert, China | Suspended aerosols | 650 - 40000 | 1 |
| Deschutter, 2022 | VMA | Internal mixture of quartz , illite and calcite using the Volume mixing approximation. | Suspended aerosols | 650 - 40000 | 1 |


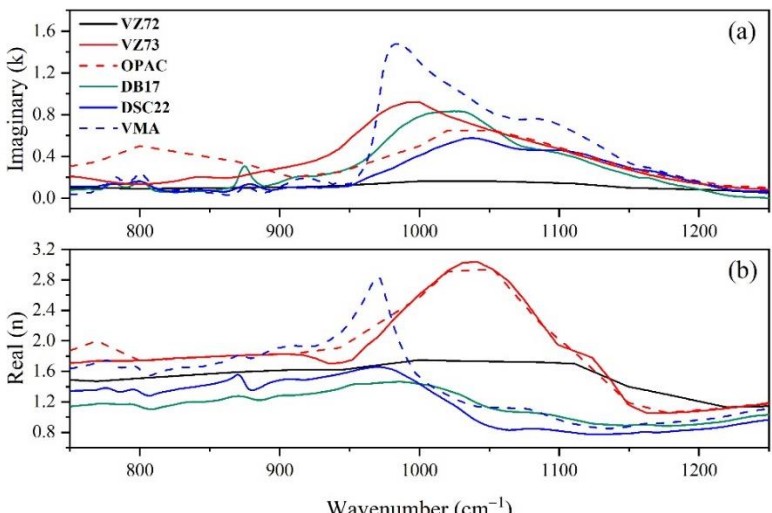

**Figure 2.** Spectral variation of the complex refractive indices of six datasets VZ72: Volz, 1972; VZ73: Volz, 1973; OPAC: Hess et al., 1998; DB17: Di Biagio et al., 2017; DSC22: Deschutter, 2022; VMA: Volume mixing approximation mixture from Deschutter, 2022.


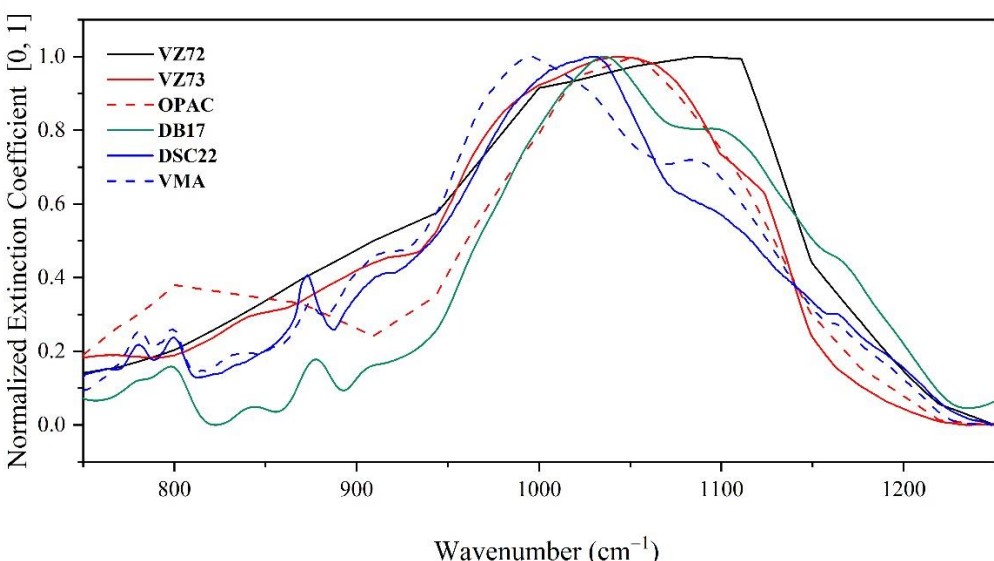

**Figure 3.** Spectral variation of the extinction coefficient using Mie theory of spherical particle of 1 **μm** of diameter on six complex refractive indices datasets VZ72: Volz, 1972; VZ73: Volz, 1973; OPAC: Hess et al., 1998; DB17: Di Biagio et al., 2017; DSC22: Deschutter, 2022; VMA: Volume mixing approximation mixture from Deschutter, 2022.


**Table 2.** Non-retrieved parameter a priori and uncertainty references.

| Non-retrieved parameters | A priori value reference | A priori uncertainty | Uncertainty reference |
|---|---|---|---|
| $H_2O$ | UWYO database | 10% | Clerbaux et al., 2007 |
| CO2 | CAMS | 1% | Engelen & Stephens, 2004 |
| O3 | WOUDC database | 5% | Boynard, et al., 2016 |
| CH4 | CAMS | 5% | De Watcher et al., 2017 |
| N2O | Mid-latitude winter standard | 100% | - |
| CFC-11 | Mid-latitude winter standard | 100% | - |
| CFC-12 | Mid-latitude winter standard | 100% | - |
| Surface Temperature | IASI l2 Product (EUMETSAT) | 1K | Pougatchev et al., 2009 |
| Surface emissivity | Zhou et al., 2014 + Alalam et al, 2022 | 2% | Capelle et al., 2012 |


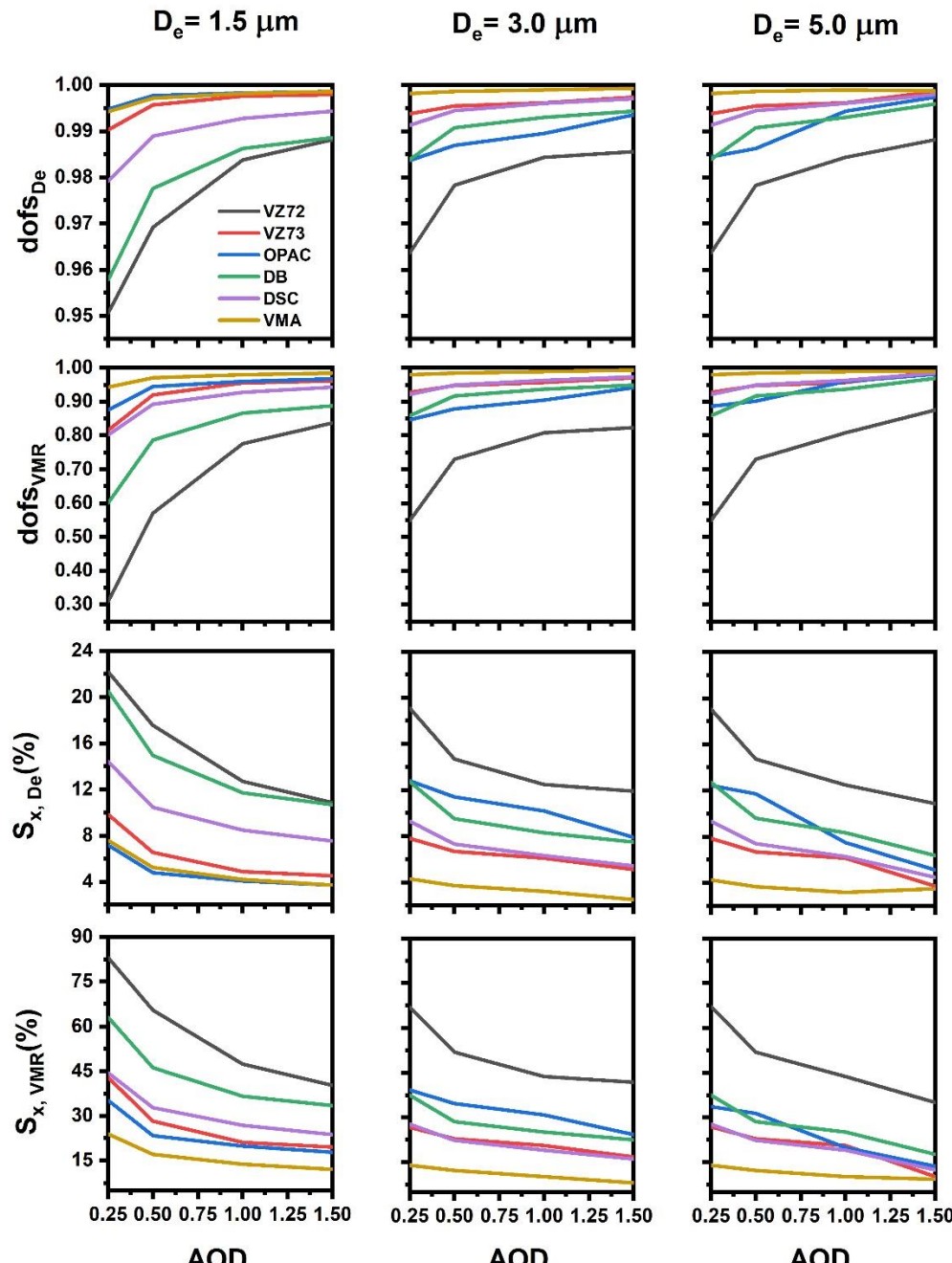

**Figure 4.** The dofs and total error $S_x$ (in %) of the state vector parameters as function of the AOD for each literature complex refractive index. $D_e$ is the effective particle diameter, and VMR is the volume mixing ratio. VZ72: Volz, 1972; VZ73: Volz, 1973; OPAC: Hess et al., 1998; DB17: Di Biagio et al., 2017; DSC22: Deschutter, 2022; VMA: Volume mixing approximation mixture from Deschutter, 2022.

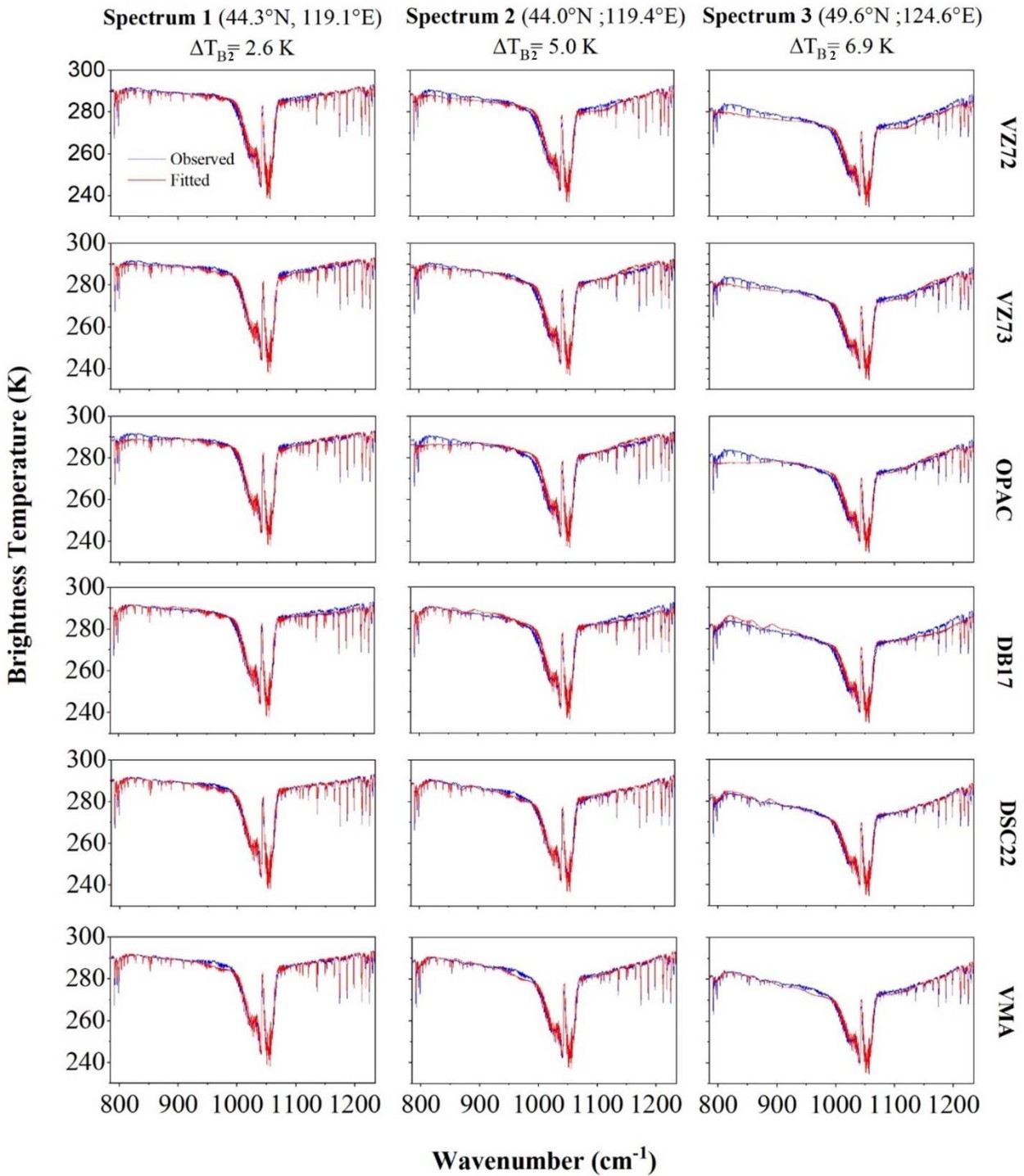

**Figure 5.** Retrieval of IASI spectral examples from 4 May 2017 using six CRI datasets: VZ72: Volz, 1972; VZ73: Volz, 1973; OPAC: Hess et al., 1998; DB17: Di Biagio et al., 2017; DSC22: Deschutter, 2022; VMA: Volume mixing approximation mixture from Deschutter, 2022.


**Table 3.** Retrieved parameters using ARAHMIS applied on three spectra using different complex refractive indices. VZ72: Volz, 1972; VZ73: Volz, 1973; OPAC: Hess et al., 1998; DB17: Di Biagio et al., 2017; DSC22: Deschutter, 2022; VMA: Volume mixing approximation mixture
from Deschutter, 2022.

| | Spectrum | VZ72 | VZ73 | OPAC | DB17 | DSC22 | VMA |
|---|---|---|---|---|---|---|---|
| **Effective diameter (μm)** | 1 | 4.5 | 3.6 | 4.9 | 4.8 | 3.4 | 3.5 |
| | 2 | 4 | 2.7 | 4.4 | 5.1 | 3.6 | 2.9 |
| | 3 | 4.2 | 3.1 | 4.3 | 5.7 | 2.9 | 3.1 |
| **VMR (in ppm)** | 1 | $4.5 \times 10^{-12}$ | $6.0 \times 10^{-12}$ | $5.9 \times 10^{-12}$ | $3.7 \times 10^{-12}$ | $6.5 \times 10^{-12}$ | $6.7 \times 10^{-12}$ |
| | 2 | $9.9 \times 10^{-12}$ | $2.6 \times 10^{-11}$ | $1.2 \times 10^{-11}$ | $5.6 \times 10^{-12}$ | $1.0 \times 10^{-12}$ | $2.4 \times 10^{-11}$ |
| | 3 | $2.0 \times 10^{-11}$ | $3.0 \times 10^{-11}$ | $3.2 \times 10^{-11}$ | $7.5 \times 10^{-12}$ | $5.1 \times 10^{-11}$ | $3.1 \times 10^{-11}$ |
| **RMS (in K)** | 1 | 1.3 | 1.3 | 1.4 | 1.2 | 0.9 | 1.2 |
| | 2 | 2.1 | 1.5 | 2 | 1.5 | 1.1 | 1.3 |
| | 3 | 2.3 | 1.8 | 2.5 | 1.8 | 1.2 | 1.3 |
| **RMS (in W.m$^{-2}$.sr$^{-1}$. (cm$^{-1}$)$^{-1}$)** | 1 | $1.6 \times 10^{-3}$ | $1.6 \times 10^{-3}$ | $1.9 \times 10^{-3}$ | $1.5 \times 10^{-3}$ | $1.2 \times 10^{-3}$ | $1.5 \times 10^{-3}$ |
| | 2 | $2.2 \times 10^{-3}$ | $1.9 \times 10^{-3}$ | $2.6 \times 10^{-3}$ | $1.9 \times 10^{-3}$ | $1.4 \times 10^{-3}$ | $1.6 \times 10^{-3}$ |
| | 3 | $2.9 \times 10^{-3}$ | $2.3 \times 10^{-3}$ | $3.2 \times 10^{-3}$ | $2.4 \times 10^{-3}$ | $1.5 \times 10^{-3}$ | $1.6 \times 10^{-3}$ |

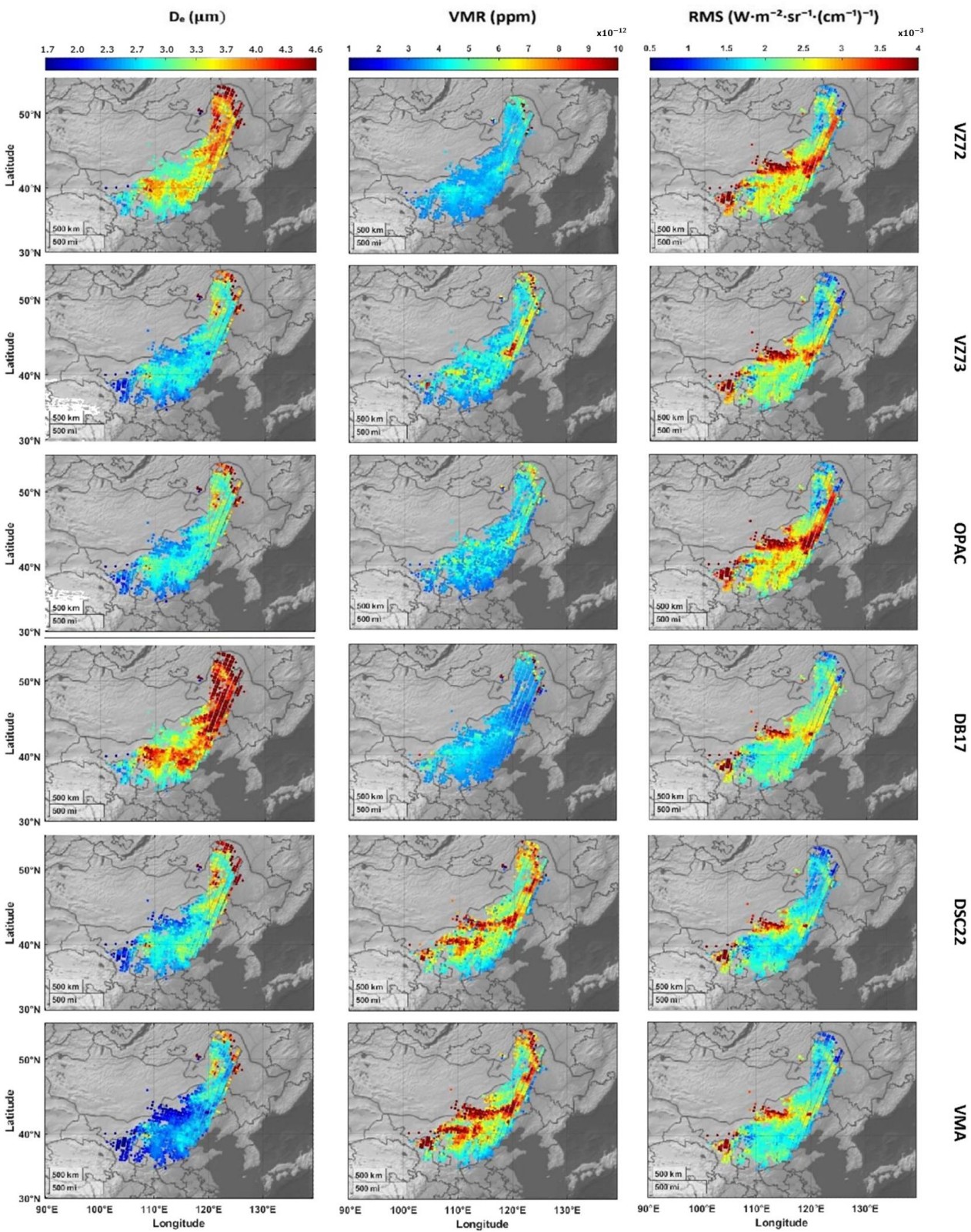

**Figure 6.** Maps of the aerosol microphysical properties ARAHMIS retrieval from the Gobi dust plume event occurred in 4 May 2017. Retrieval from IASI observations is applied on six literature complex refractive indices: VZ72: Volz, 1972; VZ73: Volz, 1973; OPAC: Hess et al., 1998; DB17: Di Biagio et al., 2017; DSC22: Deschutter, 2022; VMA: Volume mixing approximation from Deschutter, 2022.


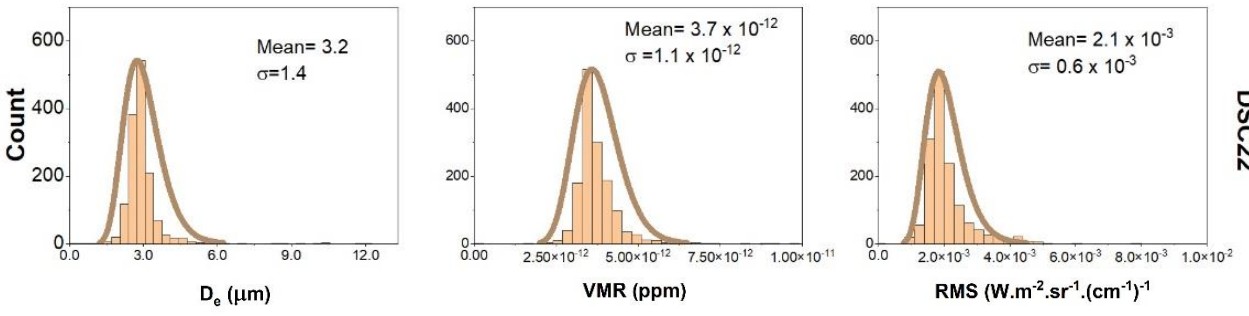

**Figure 7.** Histograms of the effective diameter, VMR and RMS from ARAHMIS retrieval of the dust plume from 4 May 2017 detected by IASI using the complex refractive index dataset DSC22: Deschutter, 2022.
