# Peer review of "The role of refractive indices in measuring mineral dust with highspectral resolution infrared satellite sounders: Application to the Gobi Desert"

_EGUsphere, 2024_

## Referee Comment (RC1)

**The role of refractive indices in measuring mineral dust with high-spectral resolution infrared satellite sounders: Application to the Gobi Desert**

The paper titled 'The Role of Refractive Indices in Measuring Mineral Dust with High-Spectral Resolution Infrared Satellite Sounders: Application to the Gobi Desert' delves into the ramifications of employing intricate refractive indices to retrieve aerosol concentration and size during episodes of intense desert dust resuspension. The authors elucidate that utilizing complex refractive indices derived from laboratory measurements enhances the accuracy of estimating desert dust concentration and size through satellite measurements (specifically, IASI). They contend that the intricate refractive index stands as a critical parameter in retrieving physical parameters, thus emphasizing the necessity of meticulous selection. Nevertheless, certain aspects of the paper lack explication that would prove beneficial to the scientific community. Moreover, some sections suffer from insufficient referencing, while one figure exhibit inaccuracies. Hence, I suggest that the authors modify the manuscript with additional precisions before publication.

Major comments:

1/ **L87-88**: In the case study, authors mentioned that the essential part of the dust aerosol suspension process is associated with strong winds and they add: « particularly during the winter and spring months ». Why is more intense during these months, are they references mentioning that?

2/**L94:** Figure 1 shows the IASI brightness difference during the Gobi dust storm. I would suggest that the authors mention the number of IASI pixels studied in that entire scene because a part of these pixels will be the ones from which the authors will want to retrieve concentration and size in the following steps of the paper.

3/**L115-154**: Regarding the CRI data section, the authors give a detailed description of the measurements of each complex refractive index (CRI) but for easier reading, I would suggest to summarize the main information in a table adding also the resolution (missing in the manuscript) for each CRI retrieval.

4/**L150:** Authors comment that the VZ72 CRI dataset was obtained « from rainout precipitation rather than desert dust ». In that way, I am wondering if this dataset is really consistent to compare with the others retrieved in dry conditions. It might be a strong impact on optical properties according to the relative humidity.

5/**L165:** Authors introduce a brand-new radiative transfer algorithm ARAHMIS but did not give any reference. Is there an existing publication describing the potential of this code or a publication mentioning the application of this code in the literature?

6/**L251:** In the Measurement error covariance matrix, authors say that "in the case of IASI instrument the SNR is set to 500". Why this typical value? IASI's instrumental noise is changing according to the frequency, is it stable in the windows 750-1250 cm$^{-1}$?

7/ Figure 6 shows the impact of the CRI used to retrieve the effective diameter ($D_e$), the volume mixing ratio (VMR) and also points out the efficiency of the retrieval through the root mean square (RMS) parameter. However, by looking to the top middle of the plume (around 45°N-115°E) the RMS of all the CRI used seems to be identical but the retrieve VMR and $D_e$ are strongly different. How these huge differences can be explained? I suggest that the RMS is a good indicator but is not sufficient alone to conclude that one CRI dataset is more accurate to another.

8/ It seems that there is a mistake in the Figure 7. The first panel ($D_e$) represents the effective diameter using DSC22 CRI but the lognormal distribution does not correspond to the Figure 6 map plot showing bigger diameter and the mean reported in the Figure 7 is not in adequation with the lognormal fit neither. At least, the mean diameter mentioned in the text L391: "3.1 µm" is not the same as the one reported in the Figure 7: "3.2 µm".

Minor comments:

1/**L27:** Missing reference for IPCC.

2/**L31-32:** A reference would be welcome to show the diversity of remote sensing instruments to capture mineral dust events.

3/**L43:** remove "of dust"

4/**L57-58:** Furthermore, Capelle et al., 2014…" the sentence is not clear for me. Could you rephrase?

5/ **L60:** add "are" in the sentence "These involved aerosol generation methods ARE more…"

6/**L285:** unit of $C_{air}$ and description of L should be added.

7/**L286**: state vector instead of "vector state"

8/**L304**: replace Fig. 3 by Figure 3

9/**L338:** same comment

10/**L359** replace Fig. 7 by Figure 7

11/Some references are cited in the text but are not included in the Bibliography section. Also, Clarisse et a., 2021 should be included after Clarisse et al., 2019. Same for Sokolik et al., 1999 & 1998.

---

## Author Response (AR1)

**General response to reviewers:**

Dear Reviewers,

Thank you for your detailed review of our manuscript titled "The role of refractive indices in measuring mineral dust with high-spectral resolution infrared satellite sounders: Application to the Gobi Desert" .We have carefully considered all your comments and suggestions. Below, we provide a point-by-point response, outlining the revisions made in response to your feedback. Most of the references cited in this response can be found in the reviewed article; only two references are explicitly cited here for clarity.

**Response to Reviewer #1:**

- **Major comments:**

***Comment 1:*** *L87-88: In the case study, authors mentioned that the essential part of the dust aerosol suspension process is associated with strong winds and they add: « particularly during the winter and spring months ». Why is more intense during these months, are they references mentioning that?*

**Response 1:** The region experiences strong winds during these seasons, especially due to the East Asian Monsoon system. These winds, often associated with cold fronts from Siberia and Mongolia. The significant temperature differences between cold Siberian air masses and warmer air to the south create strong pressure gradients, leading to high wind speeds that lift and transport dust particles in these two seasons. Wang et al., 2004 stated that in his overview of dust storms in China. The reference and a sentence are added to the revised manuscript.

***Comment 2:*** *L94: Figure 1 shows the IASI brightness difference during the Gobi dust storm. I would suggest that the authors mention the number of IASI pixels studied in that entire scene because a part of these pixels will be the ones from which the authors will want to retrieve concentration and size in the following steps of the paper.*

**Response 2:** The number of pixels used in this case study was mentioned in line 383.

***Comment 3:*** *L115-154: Regarding the CRI data section, the authors give a detailed description of the measurements of each complex refractive index (CRI) but for easier reading, I would suggest to summarize the main information in a table adding also the resolution (missing in the manuscript) for each CRI retrieval.*

**Response 3:** Thank you for the suggestion. We have created a table that summarizes the main information regarding the CRI data, including the resolution for each set. This table is now included in the revised manuscript (**Table 1.**). The spectral resolutions were also added to the text in the **section 3.2**.

***Comment 4:*** *L150: Authors comment that the VZ72 CRI dataset was obtained « from rainout precipitation rather than desert dust ». In that way, I am wondering if this dataset is really consistent to compare with the others retrieved in dry conditions. It might be a strong impact on optical properties according to the relative humidity.*

**Response 4:** Rainout precipitation contains dust particles which were transported and then deposited through the 'wet deposition'. In our case, this dataset is considered because dust particles were transported on thousands of kilometres. In addition, we chose this dataset since it has been widely recognized and employed in numerous studies, providing a robust framework for comparative analysis. The argument was added the revised manuscript.

***Comment 5:*** *L165: Authors introduce a brand-new radiative transfer algorithm ARAHMIS but did not give any reference. Is there an existing publication describing the potential of this code or a publication mentioning the application of this code in the literature?*

**Response 5:** ARAHMIS is a line-by-line radiative transfer code developed at LOA, commonly used as a reference code for preparing space missions such as IASI-NG, HYSP, and MicroCarb. Although it has not yet been the subject of a specific publication, it is mentioned in the following article: El Kattar, M.-T., Auriol, F., and Herbin, H. (2020). Instrumental characteristics and Greenhouse gases measurement capabilities of the Compact High-spectral Resolution Infrared Spectrometer: CHRIS. *Atmospheric Measurement Techniques*. https://doi.org/10.5194/amt-13-3769-2020. The details were added to the revised manuscript.

***Comment 6:*** *L251: In the Measurement error covariance matrix, authors say that "in the case of IASI instrument the SNR is set to 500". Why this typical value? IASI's instrumental noise is changing according to the frequency, is it stable in the windows 750-1250 cm⁻¹?*

**Response 6:** This value represents the equivalent noise level of the IASI instrument in the mid-infrared range between 660-1220 cm⁻¹. As shown in the figure below from Clerbaux et al., 2009, as established from a set of representative spectra, spanning a range of latitude, the noise is stable within this spectral range and has a value of approximately $2\times10^{-6}$ W/(m² sr m⁻¹).

[Figure]

***Comment 7:*** *Figure 6 shows the impact of the CRI used to retrieve the effective diameter (De), the volume mixing ratio (VMR) and also points out the efficiency of the retrieval through the root mean square (RMS) parameter. However, by looking to the top middle of the plume (around 45°N-115°E) the RMS of all the CRI used seems to be identical but the retrieve VMR and De are strongly different. How these huge differences can be explained? I suggest that the RMS is a good indicator but is not sufficient alone to conclude that one CRI dataset is more accurate to another.*

**Response 7:** The RMS values of DB17, DSC22, and VMA are significantly better than those of VZ72, VZ73, and OPAC, highlighting improvements in retrieving CRIs from resuspended particles, particularly in reproducing IASI detections. The similarity between DSC22 and VMA RMS values shows consistency between the CRIs obtained by Deschutter, 2022 and the mineralogical composition reported by Alalam et al., 2022. However, the RMS values for DB17 are consistently lower, likely due to variations in mineral compositions and the influence of humidity. DB17 reported that small amounts of water vapor and $CO_2$ contaminated the dust spectra below 7 μm. In contrast, DSC22 and VMA were obtained from the PC2A platform, which uses a nitrogen purge to reduce water vapor and $CO_2$ content. This difference could account for the observed variations. We agree also that while the RMS is a good indicator for reproducing IASI observations, results should ideally be validated with other measurement types. These information are now included in the revised manuscript.

***Comment 8:*** *It seems that there is a mistake in the Figure 7. The first panel (De) represents the effective diameter using DSC22 CRI but the lognormal distribution does not correspond to the Figure 6 map plot showing bigger diameter and the mean reported in the Figure 7 is not in adequation with the lognormal fit neither. At least, the mean diameter mentioned in the text L391: "3.1 μm" is not the same as the one reported in the Figure 7: "3.2 μm".*

**Response 8:** Thank you for pointing out the mistake in **Figure 7**. It is correct that there was an inconsistency. The graphs were initially in $D_g$ and were later converted to $D_e$. The correct histogram, which corresponds to the data, is provided in the supplementary materials. We will ensure that this correction is made in the current version of the manuscript to align the figures and the reported mean diameter. The mean diameter mentioned in the text (3.1 µm) will be updated to match the value reported in **Figure 7** (3.2 µm) .

- **Minor comments:**

*Comment 1 L27: Missing reference for IPCC.*

**Response 1:** A reference for IPCC is added to the text.

*Comment 2: L31-32: A reference would be welcome to show the diversity of remote sensing instruments to capture mineral dust events.*

**Response 2:** A reference illustrating the diversity of remote sensing instruments used to capture mineral dust events is now included.

*Comment 3: L43: remove "of dust"*

**Response 3:** Phrase was reformulated.

*Comment 4: L57-58: Furthermore, Capelle et al., 2014…" the sentence is not clear for me. Could you rephrase?*

**Response 4:** The sentence is now rephrased for clarity as follows: "Furthermore, Capelle et al., 2014 studied the sensitivity of the IASI brightness temperature to a change in complex refractive index comparing Volz 1972, 1973 to the "revisited" mineral dust CRI by Balkanski et al., 2007, showing high impact on the radiative transfer model."

*Comment 5: L60: add "are" in the sentence "These involved aerosol generation methods ARE more…"*

**Response 5:** The sentence is corrected.

*Comment 6: L285: unit of $C_{air}$ and description of L should be added.*

**Response 6:** The information were added $C_{air}$ in ppm and L is the layer thickness which was fixed by 1 km as validated by CALIPSO lidar data.

*Comment 7: L286: state vector instead of "vector state"*

**Response 7:** The sentence is corrected.

*Comment 8: L304: replace Fig. 3 by Figure 3*

*Comment 9: L338: same comment*

*Comment 10: L359 replace Fig. 7 by Figure 7*

**Response 8, 9, 10:** The journal requires us to mention figures in this way at the end of the sentences.

*Comment 11: Some references are cited in the text but are not included in the Bibliography section. Also, Clarisse et a., 2021 should be included after Clarisse et al., 2019. Same for Sokolik et al., 1999 & 1998.*

**Response 11:** The bibliography is reviewed to ensure all cited references are included. Clarisse et al.,2021 is added after Clarisse et al., 2019, and Sokolik et al. 1999 is included after Sokolik et al. ,1998.

**Response to Reviewer #2:**

- **Major comments:**

***Comment 1:*** *In particular, there are no details or references to the principal component analysis method briefly described in lines 111-114, also applied elsewhere (e.g., l. 189-190).*

**Response 1:** The principal component analysis (PCA) code used in this study is based on the method described by Atkinson et al., 2010. This method was improved by Pr. H. Herbin, as detailed in his Habilitation à Diriger les Recherches in 2014 (page 171). Additionally, Alalam et al., 2022, has provided a comparative analysis of IASI dust and cloud pixel selections with HIMAWARI 8 RGB images (see Section 2.4 in the referenced article). We have now included these references in the manuscript to provide the necessary details.

***Comment 2:*** *l. 94-99: the "V-shape" dust criterion is mentioned here, but a short description of the method and what is its sensitivity is needed (there is reference to a phd thesis, but a more specific reference would be preferable).*

**Response 2:** The V-shape dust criterion is well-known and has been thoroughly described in the scientific community. Sokolik et al., 1998 and Sokolik, 2002 have demonstrated that dust typically presents a clear V-shaped signature in the atmospheric window (800–1200 $cm^{-1}$). Additionally, a sensitivity study specific to IASI was conducted by Vandenbussche et al., 2013. The choice of channels for the brightness temperature difference $\Delta T_B$ depends on two primary factors:

1. Concentration Variability: The referenced PhD thesis indicates that as the concentration of dust increases, the V-shape becomes more pronounced, enhancing the sensitivity of the measurement to concentration variations within the atmospheric window. Based on this, we selected a minimum $\Delta T_B$ condition of 0.9K.
2. Gas Narrow Bands: Gases spectral lines can influence the radiance spectral values depending on their concentration in the atmosphere. To minimize the impact of these lines on the $\Delta T_B$ calculation, we empirically tested hundreds of mineral dust spectra to identify the optimal channels.

We have now included the references and information in the manuscript.

*-Does a difference between two brightness temperatures necessarily imply a v-shaped spectrum?*

In our analysis, we use not just two brightness temperatures but two differences in brightness temperature (representing two slopes of the V shape as illustrated in the figure below from Alalam Ph.D., 2022) to select the dust spectra. Specifically, the differences used are:

$$\Delta T_{B1} = T_{B,809.25} - T_{B,988} \text{ and } \Delta T_{B2} = T_{B,1191.25} - T_{B,1112}$$

While we use only one slope $\Delta T_{B2}$ in the quantification, as it is within the spectral range where surface emissivity has a significant effect on the spectrum. By considering $\Delta T_{B2}$, we ensure the surface emissivity is well corrected, as explained by Alalam et al., 2022. This information was added to the revised manuscript.

[Figure]

*-Are there specific assumptions on the spectral variability of surface properties needed?*

Yes, specific assumptions on the spectral variability of surface properties are needed. The IASI radiances, like any infrared sounder detecting in nadir, are strongly impacted by the surface emissivity variability. To address this, we use the Zhou et al., 2014 datasets and apply the land surface emissivity (LSE) method demonstrated by Alalam et al., 2022 to reduce the variability effect before the selection process. A correction factor is calculated using the Reststrahlen feature criterion, which is found in both the emissivity spectra and IASI radiances of desert surfaces. This approach helps to correct the surface emissivity effects differently for each observation. The full methodology is detailed in Alalam et al., 2022.

***Comment 3****: no verification data are presented, except for a generic comparison with measurements at two sites of the SONET network. Did the authors verify if other data are available, for example in the Skynet network (https://www.skynet-isdc.org/obs_sites.php)?*

**Response 3:** Thank you for your suggestion. The primary objective of this paper is to simulate and reconstruct the IASI spectra in the context of desert dust and to assess the impact of CRIs on the retrieval of microphysical properties. For this purpose, the RMS is a good indicator to evaluate this impact. However, we recognize the importance of verifying our results. We did explore the availability of additional data from both the Skynet network and AERONET. Unfortunately, we were unable to find any relevant data corresponding to the case study date of 4 May 2017 within the dust plume. Consequently, we relied on the available measurements from the two sites of the SONET network for our comparison. We acknowledge the importance of comprehensive verification and will consider expanding our verification dataset in future studies as more data become available. This will help to further validate our findings and enhance the robustness of our results. This information is now added to the revised manuscript.

***Comment 4****: the retrieval of the dust properties is made under several assumptions (single mode, if I understood well fixed standard deviation of the size distribution, uniform refractive index, fixed vertical distribution). Thus, although the correspondence between modelled and measured IR spectra is a relevant constraining method, the derived parameters might better be defined as radiatively equivalent aerosol model, than specific aerosol properties. The possible influence of different distribution standard deviations and vertical distributions (which has been shown to play a significant role on the satellite radiances, see e.g. Clarisse et al., 2019) should be addressed and investigated.*

**Response 4:**

The radiative transfer model used consider different assumption, therefore we must add some clarifications:

1. **Monomodal Distribution**: We considered a monomodal size distribution, consistent with the references cited in Clarisse et al., 2019. This approach is justified because, from an optical perspective in the thermal infrared, there is no significant way to distinguish between fine and coarse mode. Hence, a single median mode is used with a broader standard deviation. We have also verified this through laboratory measurements.
2. **Geometric to Effective Diameter**: Our retrievals in the state vector are based on geometric diameter, which we then convert to effective diameter by giving a fixed standard deviation. All the articles cited in Clarisse et al., 2019 use fixed sigma values (generally 2 or 2.2), which fall within the 1.75–2.25 range typically measured for dust aerosols (J. Reid, H. Jonsson, H. Maring, A. Smirnov, D. L. Savoie, S. Cliff, E. Reid, J. M. Livingston, M. M. Meier, O. Dubovik, and S.-C. Tsay, "Comparison of size and morphological measurements of coarse mode dust particles from Africa," J. Geophys. Res. 108, 8593–8620 (2003)). Additionally, we conducted simulations that show within the spectral window used, the most significant parameters are the concentration and the median diameter, while the effect of the standard deviation becomes important only beyond 1500 cm$^{-1}$. Finally, we note that for other types of particles, we might not make the same assumptions.
3. **Complex Refractive Index**: The complex refractive index is not uniform; we use the full spectral range in mid-infrared and with the finest spectral resolution available. This allows us to capture the variability across different wavelengths.
4. **Vertical Distribution**: The vertical distribution is fixed but it is based on actual CALIOP lidar detections observed during the case study, providing a realistic representation of the dust layer's vertical structure.

*Comment 5: Section 4 needs a better explanation and a clearer treatment. Also the notation is not clear. It is not clear why the AOD at 1020 nm is used here and how it is combined with determinations in the IR. Also here, all the calculations are made under the assumption that the only driving parameters are the median diameter and the concentration. How different factors affect the information content?*

The AOD value at 1020 nm is commonly used as a reference because it is a wavelength available in the Aeronet network data and also because it allows an easier comparison with climatological studies such as: Dubovik, O., Holben , B., Eck, T. F., Smirnov, A., Y. Kaufman, J., King, M. D., Tanré, D., and Slutsker, I.: Variability of absorption and optical properties of key aerosol types observed in worldwide locations, J.Atmos. Sci., 59, 590–608, 2002.

The state vector of the retrieved parameter $\mathbf{x_a}$ includes the VMR and $D_g$ , nevertheless many parameters are considered (the gases concentrations; pressure, temperature, surface temperature and emissivity). All these parameters are included in the information content analysis and the error budget (see section 3.3.2).

*Comment 6: Results shown in figure 6 might need a deeper discussion. Is there a reason for the large differences in the results obtained for DB17, DSC22 and VMA? All of these should somewhat take into account the specific mineralogy of the source. Two sets of CRI are for dry samples; might humidity play a role? Without validation data it is difficult to derive conclusions.*

**Response 6:** The largest differences appear between DB17 and the other datasets (DSC22 and VMA). As responded to reviewer 1, the humidity might play a role, but both DB17 and DSC22 CRIs are obtained from dry samples. The difference is mainly due to the variation in the imaginary part of the refractive index, which can arise from differences in the chemical composition of the samples. While we agree that ideally, validation/comparison with other types of measurements would be beneficial, the RMS remains a reliable indicator. It shows very similar values for DSC22 and VMA, and slightly less favourable results for DB17.

- **Minor comments:**

*Comment 1: The same event in May 2017 was analysed in the paper by Alalam et al., 2022. This study seems an expanded sensitivity study of the same retrieval, leading to similar conclusions. It may be useful to highlight overlapping parts, differences, and new results.*

**Response 1:** In Alalam et al., 2022 we were interested in characterizing the mineralogy of the dust plume, while in the this study we retrieve the microphysical properties and study the impact of six datasets, including one dataset (VMA) that presents a mixture from the retrieved mineralogy used in this article. The dataset show close results to DSC22 and indicates the validity of the mineralogical retrieval from Alalam et al., 2022. We highlight this link in the conclusion.

*Comment 2: Citation to references (e.g., parentheses) are not correct throughout the text.*

**Response 2:** The references were corrected in the revised version.

*Comment 3: line 8: emission should also be mentioned in addition to absorption and scattering*

**Response 3:** 'emission' was added to the abstract.

*Comment 4: line 11: reference to IASI seems to be misplaced*

**Response 4:** The sentence was revised and displaced afterward.

*Comment 5: l. 12: The sentence "this work reviews six prior ..." is misleading.*

**Response 5:** The sentence was rephrased in the abstract.

*Comment 6: l. 16: here and elsewhere, the authors should clearly define what are "old" and what "new" CRI datasets. Please, indicate on what quantity you find a decrease of total error by 30%. Reconstructed spectra should be mentioned.*

**Response 6:** We have clarified the terms "old" and "new" in the revised manuscript. The "old" CRI datasets refer to the widely used data in mineral dust applications that utilize the classical pellet method. In contrast, the "new" datasets incorporate the latest advancements in laboratory measurement techniques for aerosol generation. The total error reduction of 30% refers to the covariance matrix $S_x$, which has been specified and added to the abstract. Additionally, the reconstructed spectra are now mentioned in the revised version for further clarity in the abstract.

*Comment 7: l. 27: please add the reference to IPCC.*

**Response 7:** Reference was added.

*Comment 8: l. 39: simulate spectral fits? Please, clarify.*

**Response 8:** Sentence is rephrased in the revised version.

*Comment 9: l. 45: lack of size distribution knowledge in addition to CRI is a relevant factor.*

**Response 9:** We agree that the size distribution parameter is a critical factor. However, in this study, we focus on the CRI because it is essential for retrieving the size distribution. By improving the accuracy of the CRI, we inherently enhance the reliability of size distribution retrievals.

*Comment 10: l. 39-59: it is not clear if the authors refer to determinations and calculations in the visible or IR or both. There is a wide literature on dust characterization based on visible spectra. If they include visible spectra the discussion should be expanded.*

**Response 10:** Thank you for pointing out this ambiguity. Upon review, we realized that the paragraph was indeed misleading. Our study focuses exclusively on the IR instruments and their data. To avoid confusion, we have removed the reference to visible spectra from this section.

***Comment 11:*** *l. 69: is there e reference on the CESAM system? What is meant for "relevant atmospheric conditions"?*

**Response 11:** The CESAM system is described in Di Biagio et al., 2017. The word 'relevant atmospheric conditions' was a mistake and is replaced by 'dry conditions' in the revised manuscript.

***Comment 12:*** *l. 75: see also comment to l. 16: please, define what are the new CRI measurements.*

**Response 12:** We have clarified the terms "old" and "new" in the revised manuscript. The "old" CRI datasets refer to the widely used data in mineral dust applications that utilize the classical pellet method. In contrast, the "new" datasets incorporate the latest advancements in laboratory measurement techniques for aerosol generation.

***Comment 13:*** *l. 83-84: accuracy of mineral dust retrievals: of which properties?*

**Response 13:** The microphysical properties retrieval. Added to the revised manuscript.

***Comment 14:*** *l. 93: "most dispersion and visibility to IASI observations". Please, clarify. Is it due to clouds? To the dust amount, vertical distribution, surface conditions?*

**Response 14:** The phrase "most dispersion and visibility to IASI observations" refers specifically to the impact of cloud coverage during the dust event. On the 4th of May, the number of dust pixels was the highest, which allowed for better detection and analysis. This detail has been added to the revised manuscript.

***Comment 15:*** *l. 133: same as above: please specify what is intended for "atmospheric relevant conditions".*

**Response 15:** The word 'relevant atmospheric conditions' was a mistake and is replaced by 'dry conditions' in the revised manuscript.

***Comment 16:*** *l. 151: it seems that using soil from the correct area (i.e., a more similar mineralogy) may have a strong impact as well.*

**Response 16:** We agree that the soil collection region has a very high impact on the mineralogy therefore spectral signature (see Response 6 in Major comments). This has been added to the revised manuscript.

***Comment 17:*** *l. 168: I would suggest using "median diameter" or "geometric mean diameter" instead of geometric size diameter.*

**Response 17:** We used 'geometric mean diameter' in the revised version.

***Comment 18:*** *l. 183: is the surface temperature retrieval affected by the occurrence of dust?*

**Response 18:** Yes, the surface temperature retrieval is affected by the occurrence of dust. This is why it is corrected in the same manner as the emissivity (see Alalam et al., 2022).

***Comment 19:*** *l. 194-195: if I understand well, the dust vertical distribution is assumed to be a vertically homogeneous layer between 1.5 and 2.5 km height. As discussed above, how is the assumption on the dust vertical distribution affecting the results? The authors state that a reduction of visibility was associated with this event. This does not seem fully compatible with a dust layer above 1.5 km altitude.*

**Response 19:** The impact of the altitude assumption is primarily significant for determining the concentration of dust. For instance, if the plume altitude is overestimated, the concentration will be overestimated as well. Regarding the visibility, the mentioned reduction was specifically for Beijing, which is not used in this study and is located at a place and time that concern other dates of the dust event. We have modified this statement in the revised version of the text to clarify this point.

***Comment 20:*** *l. 246-247: is the effect of spectral stability included in SNR?  It is expected to contribute.*

**Response 20:** Yes, the effect of spectral stability is included in the SNR. In the Optimal Estimation Method (OEM), this effect has a weak influence on the retrieved parameters. However, it does interfere with the error budget.

***Comment 21:*** *l. 251: SNR set to 500: this requires a justification.*

**Response 21:**  As shown in the figure **in Response 6 to Major comments of Reviewer 1** (from Clerbaux et al., 2009), the IASI noise is stable in the spectral range This stability justifies setting the SNR to 500 in the specral range between 660-1220 cm$^{-1}$, around $2\times10^{-6}$ W/(m² sr m$^{-1}$). This explanation has been added to the revised manuscript.

***Comment 22:*** *l. 260: please, explain better why the accuracy on albedo (in which spectral range?) is related with the uncertainty on surface emissivity.*

**Response 22:** This phrase was removed to remove the confusion and ensure clarity in the context of this study.

***Comment 23:*** *l. 283: please, clarify what is the range of values needed to avoid saturation and to avoid loosing sensitivity.*

**Response 22:** In the infrared spectrum, the sensitivity to dust detection is influenced by the thermal contrast between the surface temperature and the dust layer temperature. This sensitivity varies with the altitude of the dust layer relative to the surface; lower altitudes result in reduced sensitivity due to the greater distance from the satellite, whereas higher altitudes increase sensitivity and allow for higher AOD until saturation occurs due to higher thermal contrast.

In our case study, the dust plume is at an average altitude of 2 km with a layer thickness of 1 km. We conducted simulations to determine the limit of sensitivity for various AOD values (equivalent at 1020 nm):

- AOD of 0.05 (with a $\Delta T_{B2} = 1$ K, close to our condition of $\Delta T_{B2} > 0.9$ K ).
- AOD of 0.2 (with a $\Delta T_{B2} = 4.8$ K ).
- AOD of 2, where the spectrum reaches saturation and emission rays occur.

The details were added in the revised manuscript. The figure below illustrates the brightness temperature as a function of wavenumbers for these AOD values. It demonstrates how sensitivity and saturation vary across the spectrum:

[Figure]

**Comment 24:** *l. 294: please, for clarity, use percent (used elsewhere) instead of fractional values for Sx.*

**Response 26:** The values were corrected.

**Comment 25:** *l. 294-295: "The relative behavior of the CRIs ...": the sentence is unclear.*

**Response 25:** The sentence was rephrased.

**Comment 26:** *l. 313-314: "... is by tenth ...": sentence unclear.*

**Response 26:** The sentence was rephrased.

**Comment 27:** *l. 324-325: The sentence needs clarification. In my opinion, you can not use RMS on spectra as a measure of uncertainty on CRI. What would be the RMS corresponding to the measured spectra noise level (SNR=500)? This might give a reference value.*

**Response 27:** We remind that the RMS equation is given by:

$$RMS \ = \ \sqrt{\Sigma(y_i \ - \ F(x_i))^2 \ / \ n}$$

where $y_i$ is the IASI observation, $F(x_i)$ is the reproduced spectrum and $n$ is the number of the IASI spectral channels of 1804.

In our study, the RMS is not used to quantify the uncertainty on the CRI; instead, it is used to evaluate the capability to reproduce the IASI observations using different CRI datasets. The radiometric noise for the IASI instrument is approximately $2\times10^{-4}$ W·m$^{-2}$·sr$^{-1}$·(cm$^{-1}$)$^{-1}$ in the mid-infrared region. Our mean RMS value of $2\times10^{-3}$ W·m$^{-2}$·sr$^{-1}$·(cm$^{-1}$)$^{-1}$ for 1447 pixels is comparable to this noise level. Given the complexity of our radiative transfer model, which includes numerous non-retrieved parameters such as temperature, pressure, and gases concentrations, a mean RMS of $2\times10^{-3}$ W·m$^{-2}$·sr$^{-1}$·(cm$^{-1}$)$^{-1}$ is within an acceptable range. This RMS value indicates that despite the challenges posed by non-retrieved parameters, our model is reasonably capable of reproducing the IASI observations. In the future, further improvements can be targeted to refine the treatment of non-retrieved parameters. u

We have added this clarification to the text to provide a more comprehensive understanding of the RMS usage and its implications in our study.

**Comment 28:** *l. 325-326. "We selected spectra with large spectral features, whose simulation presents significant challenges and thereby provides a rigorous test for the CRIs" please, clarify. The sentence seems contradictory (you need large spectral features for detection).*

**Response 28:** We agree that the original sentence was misleading. We have rephrased it in the revised manuscript for clarity. When selecting spectra for this study, we focused on those with pronounced spectral features because they provide a rigorous test for evaluating the CRIs. As we approach saturation, the spectra increasingly reflect thermal emission from the aerosol layer, leading to a loss of sensitivity due to reduced spectral variation. This makes it more challenging to accurately reproduce the spectra. Table 3 demonstrates that as aerosol loading increases, our ability to accurately reproduce the IASI spectra decreases, as given by the increasing RMS values. We have modified the text to reflect these clarifications.

**Comment 29:** *l. 328-330: looking at fig. 5, it seems that for all cases and all CRI datasets there are spectral regions with model/measurement differences which probably exceed SNR=500 on measurements. May this provide useful suggestions on lacking knowledge on dust properties?*

**Response 29:** A SNR of 500 corresponds to an RMS of $2 \times 10^{-4}$ $W \cdot m^{-2} \cdot sr^{-1} \cdot (cm^{-1})^{-1}$ (Check **Response 27**). In comparison, the cases of VZ72, VZ73, and OPAC are less accurate then DB17, DSC22 and VMA. Nevertheless, there are two regions in the neighbouring of (38°N; 104°E and 43°N; 112°E) where the RMS is slightly higher for all CRIs. This corresponds to finer size distributions for which the thermal IR is less sensitive, and this is confirmed by the diameter retrieval for all CRIs. This has been added to the revised version.

**Comment 30:** *l. 330-331: although this might be the case, you might have compensating effects among different variables.*

**Response 30:** We acknowledge that compensating effects among different variables can occur. This is particularly true for fine particles, to which the IR is less sensitive, leading to higher VM for these values. To quantify this effect accurately, it is essential to have a sufficient number of ground-based measurements that are statistically representative of the dust plume location to validate this compensating effect. However, in this case study, the availability of ground-based measurements is very limited. This detail was added in the conclusion.

**Comment 31:** *l. 339: is there an optimal range of absorption/extinction for detection? Is the largest extinction due to too much absorption (ie., imaginary part of CRI)?*

**Response 31:** To detect dust, the optimal range of aerosol optical depth (AOD) is above 0.05 at 1020 nm. However, to avoid saturation, which reduces detectability due to the dominant emission effect, AOD should be below 2 at 1020 nm as explained in **Response 22**. The largest extinction is influenced by both the absorption due to the imaginary part of the refractive index and the microphysical properties (diameter, VMR) coupled with it.

**Comment 32:** *l. 352: what was local time?*

**Response 32:** Between 10 am and 12 pm in Beijing local time. Added to the revised manuscript.

**Comment 33:** *l. 354: "The mean value indicates the range of each parameter": please, clarify.*

**Response 33:** The sentence was rephrased.